# An ancient family of lytic polysaccharide monooxygenases with roles in arthropod development and biomass digestion

Federico Sabbadin[1], Glyn R. Hemsworth [2,3], Luisa Ciano [4], Bernard Henrissat[5,6,7], Paul Dupree [8], Theodora Tryfona[8], Rita D.S. Marques[8], Sean T. Sweeney [9], Katrin Besser[1], Luisa Elias[1], Giovanna Pesante[1], Yi Li[1], Adam A. Dowle[10], Rachel Bates[10], Leonardo D. Gomez[1], Rachael Simister[1], Gideon J. Davies [4], Paul H. Walton [4], Neil C. Bruce[1] & Simon J. McQueen-Mason[1]

*Thermobia domestica* belongs to an ancient group of insects and has a remarkable ability to digest crystalline cellulose without microbial assistance. By investigating the digestive proteome of *Thermobia*, we have identified over 20 members of an uncharacterized family of lytic polysaccharide monooxygenases (LPMOs). We show that this LPMO family spans across several clades of the Tree of Life, is of ancient origin, and was recruited by early arthropods with possible roles in remodeling endogenous chitin scaffolds during development and metamorphosis. Based on our in-depth characterization of *Thermobia*'s LPMOs, we propose that diversification of these enzymes toward cellulose digestion might have endowed ancestral insects with an effective biochemical apparatus for biomass degradation, allowing the early colonization of land during the Paleozoic Era. The vital role of LPMOs in modern agricultural pests and disease vectors offers new opportunities to help tackle global challenges in food security and the control of infectious diseases.

[1] Centre for Novel Agricultural Products, Department of Biology, University of York, York YO10 5DD, UK. [2] School of Molecular and Cellular Biology, Faculty of Biological Sciences, University of Leeds, Leeds LS2 9JT, UK. [3] Astbury Centre for Structural Molecular Biology, University of Leeds, Leeds LS2 9JT, UK. [4] Department of Chemistry, University of York, York YO10 5DD, UK. [5] Architecture et Fonction des Macromolécules Biologiques (AFMB), UMR 7257 CNRS, Université Aix-Marseille, 163 Avenue de Luminy, 13288 Marseille, France. [6] INRA, USC 1408 AFMB, 13288 Marseille, France. [7] Department of Biological Sciences, King Abdulaziz University, Jeddah 21589, Saudi Arabia. [8] Department of Biochemistry, University of Cambridge, Cambridge CB2 1QW, UK. [9] Department of Biology, University of York, York YO10 5DD, UK. [10] Bioscience Technology Facility, Department of Biology, University of York, York YO10 5DD, UK. Correspondence and requests for materials should be addressed to S.J.M-M. (email: simon.mcqueenmason@york.ac.uk)

Cellulose and chitin are the most abundant polysaccharides on earth and provide the structural load-bearing framework in the cell walls of many organisms (plants, fungi, arthropods). The benefits of these paracrystalline polysaccharides are tensile strength similar to steel, inherent rigidity, and high chemical stability. The enzymatic and physiological mechanisms underpinning cellulose and chitin metabolism in simple and complex organisms have become of increasing interest as powerful new tools for a wide range of industrial, agrochemical, and medical applications. In recent years, understanding of biological biomass degradation has been overturned by the discovery and characterization of copper-containing[1] lytic polysaccharide monooxygenases (LPMOs)[2]. LPMOs are now known to play a pivotal role in the breakdown of polysaccharides such as cellulose and chitin[1–3], by catalyzing the oxidative, as opposed to hydrolytic, cleavage of glycosidic bonds. In this way, the initial chemical and physical recalcitrance of the polysaccharide is overcome, thereby making the substrate tractable to hydrolases[1, 2, 4, 5]. Indeed, such is the effect of LPMOs that they are now included in commercial biomass saccharification cocktails, driving large advances in the environmental and commercial sustainabilities of second generation biorefineries[6]. Until now, only LPMOs from bacterial, fungal, or viral genomes[7] have been characterized, with a predominant interest in their industrial applications toward bioethanol production.

Commonly known as the firebrat, *Thermobia domestica* (Fig. 1a) is a detritivorous insect related to the silverfish and belonging to the order of *Zygentoma*, one of the most primitive groups of insects that appeared on land during the Devonian Period (420 million years ago)[8]. These animals can efficiently digest crystalline cellulose at rates comparable to cows and termites, but unlike these animals, digestion in firebrats is accomplished without microbial assistance, thus making the endogenous proteins responsible for biomass utilization of significant importance to both evolutionary entomology and industrial biotechnology[9–13]. Here, we report the investigation into the digestive enzymes from *T. domestica*, which we show include members of an uncharacterized family of endogenous LPMOs. Phylogenetic analysis reveals that this family is widespread across Phyla not previously known to possess LPMOs, including algae, oomycetes, and complex animals, and is active on both cellulose and chitin. In-depth biochemical, structural, and spectroscopic data, gene expression patterns, and gene suppression phenotypes suggest that these ancient LPMOs play crucial roles in arthropod development and food digestion, and represent a new range of tools to help tackle major challenges in agriculture and public health.

## Results

**Shotgun proteomics.** To investigate the digestive enzymes produced by *T. domestica*, we grew batches of individuals on different carbon sources and isolated the content of the crop, which represents the largest organ of the foregut (Fig. 1b). Microscopic analysis of the crop content from animals grown on microcrystalline cellulose (Avicel) revealed that the particle size of cellulose was markedly reduced (Fig. 1c, d). High-performance anion-exchange chromatography (HPAEC) analysis of the fluids of the crop from animals grown on Avicel showed a dominant peak corresponding to glucose, indicating that crystalline cellulose had been broken down to its monomeric unit (Fig. 1e). Agar plate and in vitro activity assays carried out with the soluble proteins extracted from the crop revealed the ability to breakdown a wide range of complex polysaccharides normally found in plant biomass, including glucans, mannans, and xylans (Supplementary Figs. 1 and 2), suggesting the presence of a complex

enzymatic cocktail. We sought to identify the enzymes responsible for the breakdown of polysaccharides in *Thermobia* by performing shotgun proteomic analysis of the gut content from animals that had been fed oat flour, pulverized wheat straw, filter paper, or crystalline cellulose (Avicel) as the main carbon source. Our analysis revealed that the gut proteome of *Thermobia* is dominated by recognizable carbohydrate active enzymes (CAZymes, Fig. 1f, Supplementary Datas 1–4) that make up around half of the total protein content. Virtually all these sequences have best BlastX matches with genomic orthologues from insects, with the exception of a small number of putative bacterial glycoside hydrolases belonging to family 30 (GH30) (Fig. 1f, Supplementary Datas 1–4). Amongst the gut luminal proteins was a group of unknown function that showed increased abundance in animals fed on crystalline cellulose-enriched diets, reaching 20% of the gut CAZymes (Fig. 1f) in animals grown on Avicel. Our proteomic data indicated the presence of 21 such proteins with corresponding ESTs in the *T. domestica* transcriptome.

**Phylogeny and sequence analysis.** Interrogation of public databases (BlastP vs. NCBI nr databases), using the above mentioned 21 uncharacterized proteins from *Thermobia* as queries, identified hundreds of orthologous sequences in the annotated genomes of marine and terrestrial invertebrates (crustaceans, molluscs, insects, millipedes, and spiders) and beyond the animal kingdom into disparate groups of algae and oomycetes (Fig. 2). All sequences shared distant sequence similarity (between 20% and 30% amino acid identity) to LPMOs (as confirmed by Pfam search), most evident in a conserved N-terminal histidine brace that coordinates the active site copper[1] (Supplementary Fig. 3).

LPMOs are classified in the carbohydrate active enzymes (CAZy) database (www.cazy.org)[14] as Auxiliary Activity (AA) enzyme families AA9-AA11, AA13, and AA14. We selected 240 non-fragmentary sequences orthologous to *Thermobia*'s putative LPMOs, and searched them against the Hidden Markov models (HMMs) of LPMO families AA9-AA11, AA13, and AA14. This analysis did not find any significant hits. However, the same queries showed excellent hits ($<E^{-41}$, Supplementary Data 5) against a newly computed HMM, built from their multiple sequence alignment. The high statistical significance of the new HMM, and the lack of significant hits against HMMs of previously characterized LPMO families, allow us to rigorously define a new family, which will appear as AA15 in the CAZy database.

Although all LPMO families, until now, stem from bacterial, fungal, or viral genomes[7], putative endogenous LPMO sequences have previously been identified in some insect species using bioinformatics[15]. However the complex phylogeny, biochemical properties and role of those enzymes have not been investigated. The identification of over twenty such LPMOs in the digestive system of *Thermobia* revealed an unexpected diversity within a single organism and provided an important clue about the function of these enzymes in insects. Our analysis also showed that the AA15 family is widespread among crustaceans, molluscs, chelicerates, algae, and oomycetes, none of which has previously been known to possess LPMOs (Fig. 2). Interestingly, while sequences in *T. domestica* only correspond to the LPMO catalytic domain, about a third of the members of the family identified in other species harbor a C-terminal extension. Most of these C-terminal extensions can be assigned to various carbohydrate-binding domain (CBM) families based on amino acid sequence relatedness (for classification of CBMs see CAZy database). The fusion of AA15 members to CBM1 (cellulose-specific) and CBM14 (chitin-specific) domains suggests that this LPMO family

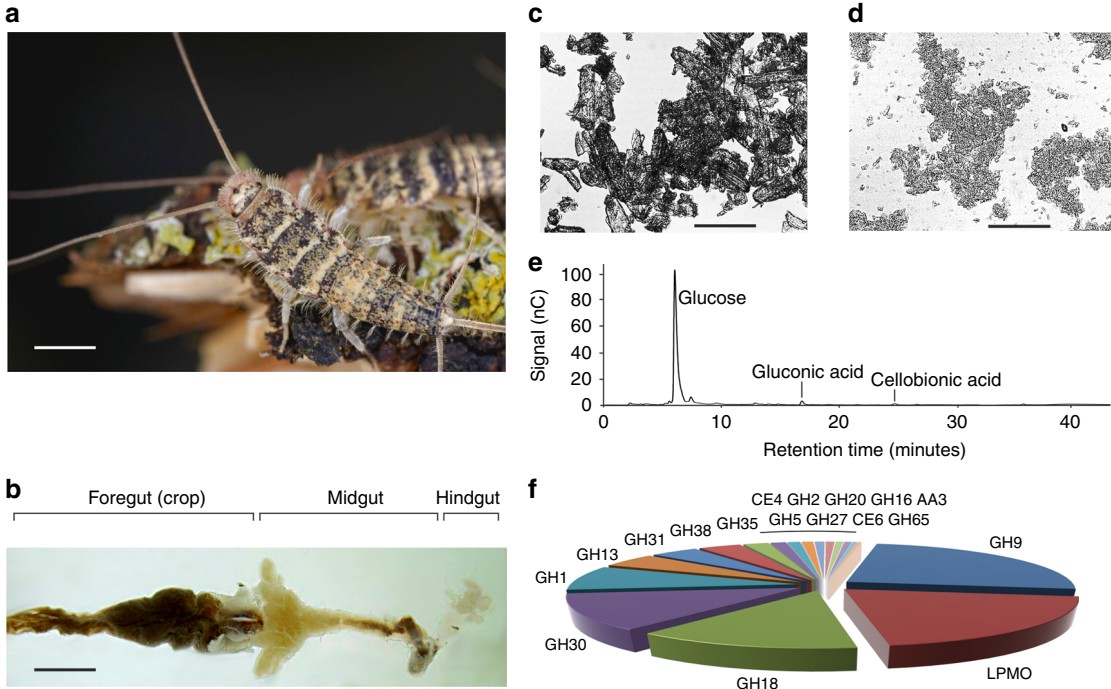

**Fig. 1** Discovery of the AA15 LPMO family in *T. domestica*. (**a**) Photograph of live specimens of *T. domestica* in their natural environment. Scale bar, 300 μm. (**b**) Dissected gut of *T. domestica*. The crop represents the largest portion of the foregut and the organ where food particles and digestive enzymes accumulate. Scale bar, 100 μm. (**c**) Microscopic image of Avicel (microcrystalline cellulose). Average particle size is ~50 μm. Scale bar, 30 μm. (**d**) Microscopic image of food pellet collected from the crop of *T. domestica* fed on Avicel. Particle size is greatly reduced to ~5 μm. Scale bar, 30 μm. (**e**) HPAEC analysis of soluble extract isolated from the crop of *T. domestica* grown on Avicel. One dominant peak corresponding to glucose is clearly visible, plus minor peaks for gluconic acid and cellobionic acid. The identity of the peaks was determined by analyzing commercial standards. (**f**) Pie chart summary of the CAZymes identified in the crop of *T. domestica* grown on Avicel. Abundance values for the various families were calculated as molar percentage from emPAI values obtained from shotgun proteomics data (GH9 24.6%, LPMO 20.2%, GH18 13.2%, GH30 12.3%, GH1 8.4%, GH13 4.7%, GH31 4.0%, GH38 3.6%, GH35 2.1%, CE4 1.3%, GH2 1.1%, GH20 1.0%, GH16 0.8%, AA3 0.7%, GH5 0.7%, GH27 0.6%, CE6 0.4%, GH65 0.3 %). See Methods for more details

could potentially target both cellulose and chitin[16]. In addition, some of the identified AA15 LPMOs are fused to GH18 (Chlorophyta, Bacillariophyceae and tunicates) or GH19 (Oomycota, Haptophyta) domains, both classified as chitinases (Supplementary Fig. 4).

Out of 23 full-length LPMO catalytic domain encoding sequences identified in the transcriptome of *T. domestica*, peptides representing 21 were detected in significant amounts in the gut proteome. Such LPMO diversification within a single organism has been previously observed only in fungi and might indicate isoform-specific preference toward different substrates, electron donors, pH, and temperatures[17].

Protein sequence analysis revealed that all LPMOs from *T. domestica* carry a signal peptide that, once removed, allows the exposure of the conserved N-terminal catalytic histidine of the mature, secreted protein (Supplementary Fig. 3). Ten cysteine residues (potentially forming stabilizing disulfide bridges in the proteins) were found to be conserved both in the *T. domestica* LPMOs (Supplementary Fig. 3) and in the best BlastP matches found in crustaceans, molluscs, insects, and spiders. Fungal LPMOs possess an N-terminal methylated histidine but this was not observed in proteomic analyses of the LPMOs from *T. domestica* (see Methods for more details).

We performed RNA-seq analysis of the published transcriptome data for *Thermobia* and determined the relative gene expression (as Transcripts Per kilobase Million, TPM) of all assembled contigs. The analysis confirmed that most putative LPMOs in this insect are expressed at medium (10 < TPM < 100), high (100 < TPM < 1000), or very high (TPM > 1000) levels (Supplementary Table 1). In order to investigate the tissue-specific expression of the LPMO genes, we carried out RT-PCR of randomly selected sequences with cDNA derived from several tissues and observed that the LPMO genes were most highly expressed in the midgut (Supplementary Fig. 5a). We extracted genomic DNA from the legs (free of potential gut microbes) of *T. domestica* and used this as a template to amplify and sequence the full gene of one LPMO, the intron–exon architecture of which strongly supports the endogenous origin of these enzymes (Supplementary Fig. 5b, Supplementary Note 1). Analogous intron–exon gene structures were found in virtually all the AA15 orthologues identified in the published genomes of other invertebrates.

**Biochemical characterization.** The coding sequence representing one of the most abundant *T. domestica* LPMOs (contig GASN01405718.1, henceforth termed *Td*AA15A) was cloned and expressed in *Escherichia coli* with a C-terminal strep-tag. The recombinant protein was purified from the bacterial periplasm by affinity chromatography (Supplementary Fig. 5c) and characterized. Gene expression was carried out using a minimal medium devoid of metals and the purified LPMO was not bound to copper. Thermal shift analysis (Thermofluor) of purified apo-*Td*AA15A indicated a melting temperature ($T_m$) of 58.5 °C, which increased to 64 °C upon addition of excess copper and was retained after size exclusion chromatography (Supplementary Fig. 5d). Stripping copper with 10 mM EDTA lowered the $T_m$ back to 58.6 °C. These results indicate that the apo-enzyme folds correctly in the periplasm of *E. coli* and that subsequent addition

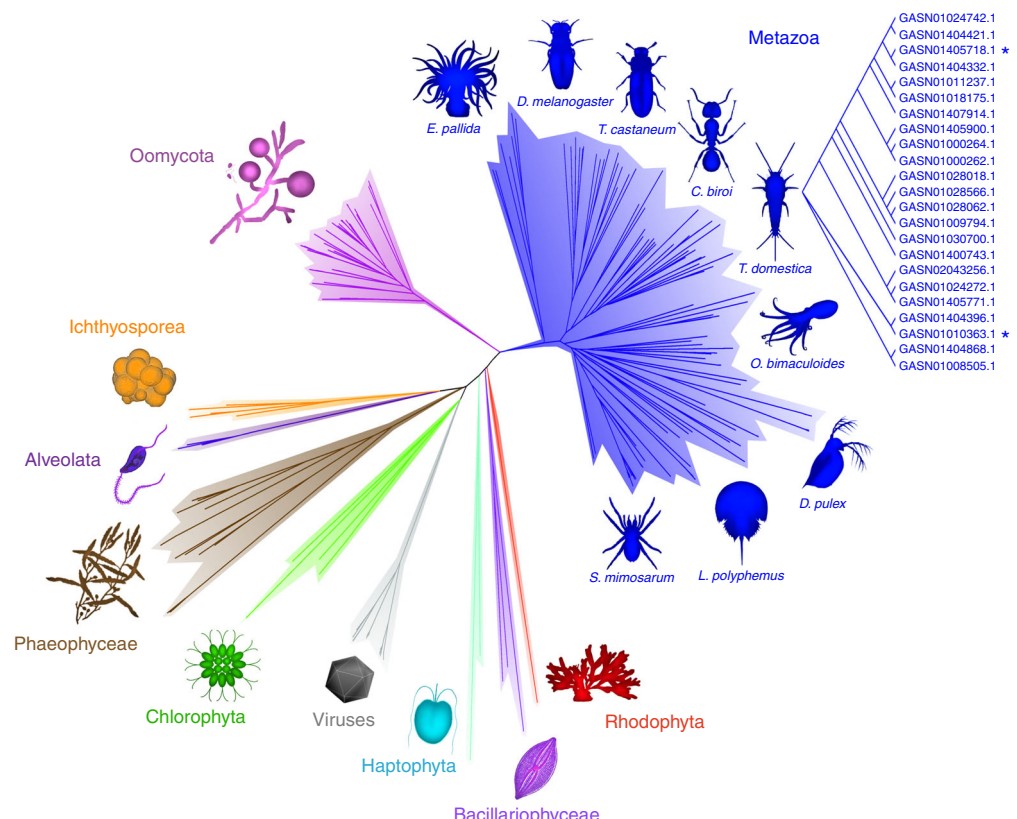

**Fig. 2** Radial phylogram of the AA15 family across Taxa. Sequences were identified in the genomes of animals (Metazoa), Oomycota, multicellular algae (Phaeophyceae, Rhodophyta), unicellular algae (Bacillariophyceae, Haptophyta, Chlorophyta, Alveolata), Ichthyosporea and viruses. The figure shows some examples of animal species (cnidaria, molluscs, insects, crustaceans, chelicerates) possessing AA15 sequences; 23 full-length LPMO domains were identified in the transcriptome of *T. domestica*. Asterisks mark the two sequences (GASN01405718.1 and GASN01010363.1, named *Td*AA15A and *Td*AA15B in this manuscript) that were successfully expressed in *Escherichia coli* and characterized. See Methods for more details

of copper increases the $T_m$ and protein stability, as observed with other LPMOs[18]. Inductively coupled plasma-mass spectrometry (ICP-MS) analysis of the reconstituted *Td*AA15A after gel filtration indicates a copper/protein ratio of $1.1 \pm 0.2$, thus suggesting saturation of the active site.

Activity assays were carried out on microcrystalline cellulose (Avicel) and β-chitin (squid pen chitin). Samples were analyzed by MALDI-TOF MS and peak masses of the reaction products compared to previously published data[1–3], revealing a predominant C1-oxidation pattern and generation of C1-aldonic acids on both substrates in presence of an external electron donor (Fig. 3a, b). Oxidized products were not detected in any of the negative controls (Supplementary Fig. 6a–d). We unambiguously confirmed C1-oxidation by MALDI-TOF MS and MS/MS analysis of the permethylated cellulose cleavage products generated by *Td*AA15A using phosphoric acid swollen cellulose (PASC) as substrate (Supplementary Fig. 6e–h). MALDI-TOF MS analysis of crude extract from activity assays carried out with Cu-loaded *Td*AA15A in the presence of 10 mM EDTA failed to detect the release of both native and oxidized oligosaccharides (Supplementary Fig. 7a, b), indicating that the apo-enzyme is not active and that copper is essential for activity. By quantifying product formation via HPAEC, we identified gallic acid as the most effective reductant (Supplementary Fig. 8a) and used it in all subsequent synergy experiments. These were carried out with commercially relevant cellulases and chitinases and the released products were quantified by HPAEC. Reactions containing either *Td*AA15A or the glycoside hydrolase alone released small amounts of oligosaccharides from cellulose or chitin, while co-incubation reactions containing both enzymes dramatically

increased the yield. The LPMO synergized with hydrolases belonging to GH6 (cellobiohydrolase), GH7 (endoglucanase), GH9 (endoglucanase), GH1 (β-glucosidase), and GH18 (endochitinase) families. Such boosting was further enhanced by addition of gallic acid as electron donor (Fig. 3c, d; Supplementary Fig. 8b–i).

**Structural characterization of *Td*AA15A.** We determined the crystal structure of *Td*AA15A to 1.1 Å resolution (Supplementary Table 2), revealing the typical central β-sandwich fold of LPMOs, decorated with diverse loops and stabilized by five disulfide bonds (Fig. 4a). The Dali server was used to compare the fold more widely with other structures in the Protein Data Bank (PDB)[19] revealing that *Td*AA15A most closely resembles bacterial AA10 LPMOs with the best structural match being the *Serratia marcescens* AA10, CBP21[2, 20]. The *Td*AA15A active site is also characterized by the ubiquitous LPMO histidine brace (Fig. 4b). His1 and His91 directly coordinate the essential copper cofactor with a T-shaped geometry as observed for all copper bound LPMOs characterized to date. Additionally, the axial, non-coordinating active site residue of *Td*AA15A is a tyrosine (Tyr184) as observed in most AA9s, while the positioning of Ala89 is reminiscent of AA10s (Fig. 4c, see discussion of spectroscopy below for effects on the Cu(II) geometry). Since the enzyme was heterologously produced in *E. coli*, His1 was not methylated and therefore represented the state of the native protein from *T. domestica*, as previously revealed by our proteomics analysis of the digestive fluids, and adding an additional layer of similarity to the bacterial and virally encoded AA10s.

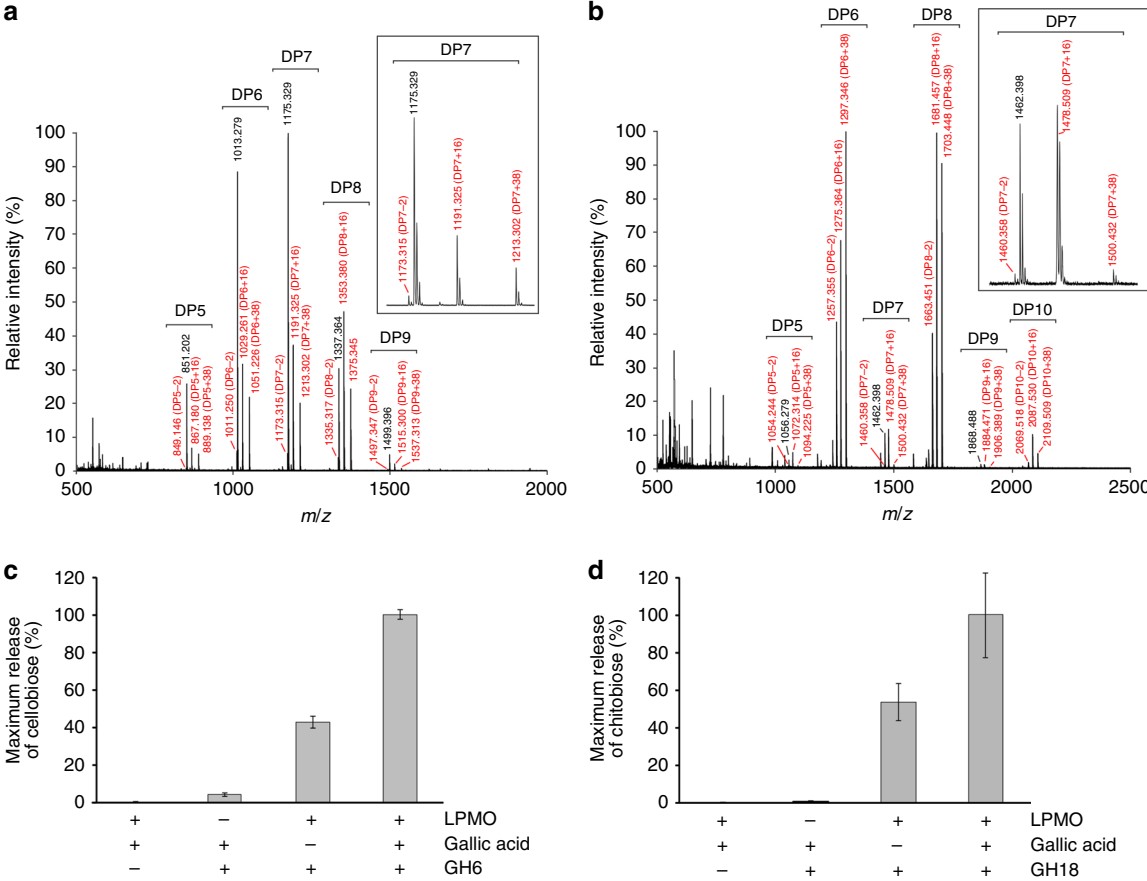

**Fig. 3** Biochemical characterization of *Td*AA15A. MALDI-TOF MS spectrum of products obtained after incubation of 4 mg mL$^{-1}$ microcrystalline cellulose (**a**) or β-chitin (**b**) with 2 μM *Td*AA15A and 4 mM gallic acid for 24 h, showing native and oxidized oligosaccharides. For both substrates, the main peaks correspond to mono- or di-sodiated adducts of C1-aldonic acids, imparting +16 or +38 *m/z* respectively, relative to the mono-sodiated unoxidized form. Smaller peaks for the mono-sodiated lactone (−2) were also identified. All oxidized species are marked in red. For chitin, the products released seem to be predominantly even-numbered oligosaccharides, implying that the enzyme can attack the crystalline structure, as previously observed for other LPMOs[2]. In **a** and **b**, 100% relative intensity represents $0.9 \times 10^4$ and $1.0 \times 10^4$ arbitrary units (a.u.), respectively. Negative control reactions carried out with substrate only, substrate plus gallic acid, and substrate plus *Td*AA15A did not generate any oxidized products (see Supplementary Fig. 6a–d). Insets are expanded mass spectra for DP7 products. **c** Relative product quantification, showing release of cellobiose from microcrystalline cellulose by a commercial GH6. The LMPO significantly boosts the activity of the GH6, and such effect is increased by addition of 1 mM gallic acid. **d** Relative product quantification, showing release of chitobiose from β-chitin by a commercial chitinase. The LPMO boosts the activity of the chitinase, and the synergy is further enhanced by the presence of 1 mM gallic acid. All boosting experiments were carried out over 3 h at 28 °C and products quantified by HPAEC. Bars indicate means (error bars: standard deviations of three replicates). The identity and quantity of each species was determined by analysis of commercial standards. See Methods for more details

While having most of the canonical features found in other LMPOs, the AA15 structure reveals an unusual β-tongue-like protrusion which links strands 8 and 9 (Fig. 4a) and forms part of the surface surrounding the active site. To investigate whether the protrusion is a modification specific to this family of LPMOs, we aligned the sequences of 214 family members identified in CAZy, including 21 from *T. domestica*, and analyzed the sequence conservation using the ConSurf[21] server (Fig. 4d). This analysis highlighted the absolute conservation at the protein active site within the family but suggests that the protrusion, while found in all *T. domestica* LPMOs, is not necessarily conserved across the whole AA15 family. Determination of its importance for mediating substrate specificity will, therefore, require further structural and biochemical characterization of other family members.

Interestingly, on opposite sides of the histidine brace and almost perfectly mirroring each other, are the co-planar aromatic rings of Tyr166 and Tyr24 (Fig. 4d), which mark the boundaries of the flat surface surrounding the active site and could be involved in substrate binding[22]. A chain of aromatic residues also forms a path through the enzyme core and could conceivably mediate electron transfer (Fig. 4e–g), as previously suggested for other AA families[3, 18, 23], possibly via one of the putative dehydrogenases[24] identified in the gut proteome of *Thermobia* (Supplementary Data 1–4).

**Spectroscopic features of *Td*AA15A**. UV-vis and Electron Paramagnetic Resonance (EPR) spectroscopies were used to further probe the copper active site of *Td*AA15A in solution. The UV-vis spectrum of the Cu(II) form of *Td*AA15A showed a broad, low-intensity signal centered around 610 nm with $\varepsilon = 75$ M$^{-1}$ cm$^{-1}$ (Supplementary Fig. 9). EPR spectroscopy revealed a complex spectrum which could be interpreted as a mixture of two different species (Fig. 4h, Supplementary Table 3). The parallel values for both species could be determined accurately (species 1, $g_z = 2.254$, $|A_z| = 525$ MHz; species 2, $g_z = 2.283$, $|A_z| = 407$ MHz) (Supplementary Table 3 and Supplementary Fig. 10), and their ratio was shown to be dependent on pH, buffer, and glycerol

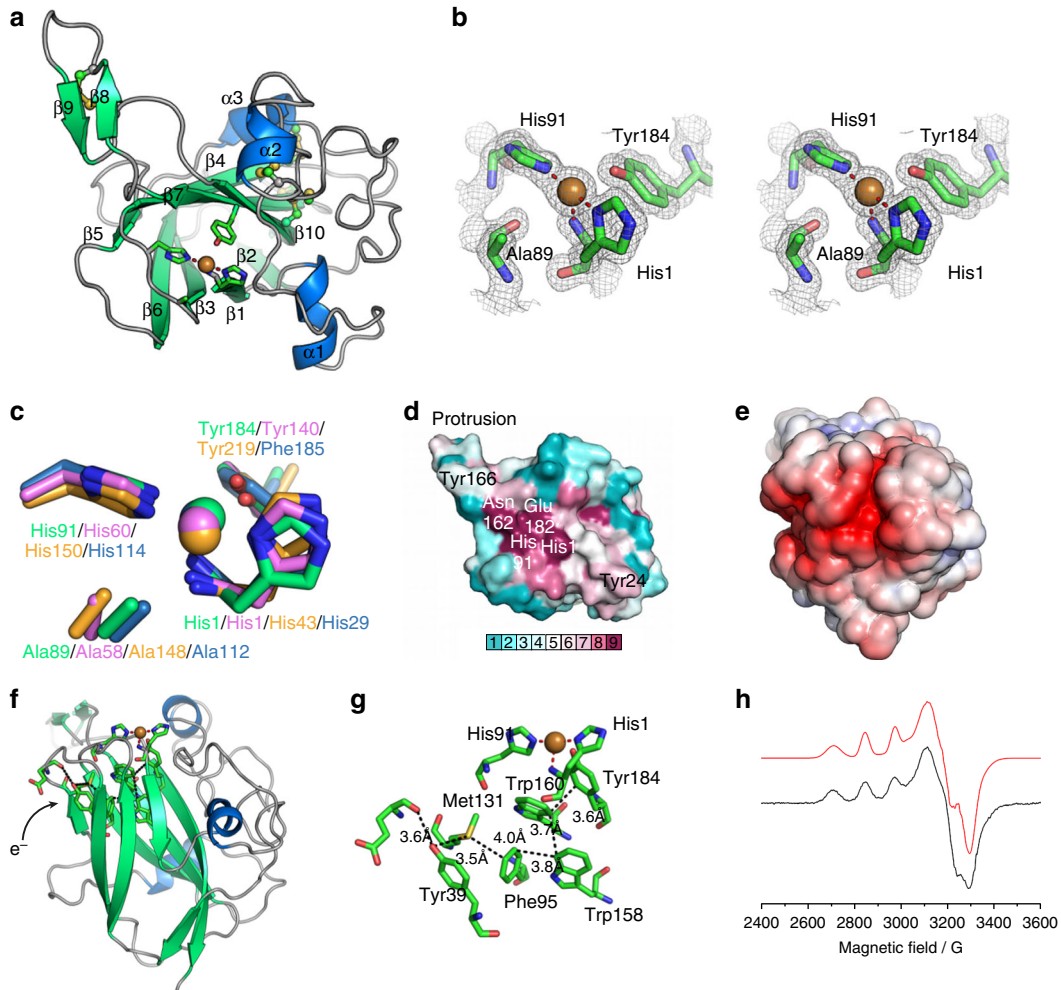

**Fig. 4** Structural and spectroscopic characterization of *Td*AA15A. **a** The overall structure of *Td*AA15A is shown colored by secondary structure. The protrusion formed by strands β8 and β9 is clearly shown extending the surface surrounding the active site. Disulfide bonds are shown in ball and stick with sulfur atoms colored yellow. **b** Stereo view of the electron density observed at the copper active site of *Td*AA15A (2mFo-Fc map contoured at 1σ). The histidine brace coordination of the copper, which is in the Cu(I) state due to photoreduction in the X-ray beam, is shown by red dashed lines. The copper ion is shown as a golden colored sphere. **c** The active site of *Td*AA15A (green) was superposed with equivalent residues from *Ao*AA11[3] (pdb 4mai, violet, a C1 specific chitin active LPMO), *Sc*LPMO10B[68] (pdb 4oy6, orange, an LPMO with both C1 and C4 oxidizing activity on cellulose, and C1 oxidizing activity on chitin), and *Ef*AA10[69] (pdb 4als, blue, a chitin active C1 specific LPMO) giving rmsds of 0.71 Å over 37 atoms, 1.13 Å over 37 atoms, and 0.59 Å over 37 atoms, respectively. The axial alanine in the chitin active *Ef*AA10 superposes closely with Ala 89 from *Td*AA15A while the equivalent alanines in *Ao*AA11 and *Sc*LPMO10B do not occupy the same position. **d** Sequence conservation analysis (ConSurf) of *Td*AA15A looking down on the active site. The surface is colored by ConSurf score according to the indicated scoring scheme. The conserved residues in the active site are labeled along with the protrusion and surface exposed tyrosines (Tyr166 and Tyr24). **e** Electrostatic surface potential of *Td*AA15A showing the large negatively charged patch (red) present on the protein surface that could be a docking site for a protein partner. The APBS plugin for PyMol was used to calculate and visualize the surface electrostatic potential at ±5 KBT/e. **f** Cartoon representation of *Td*AA15A in the same orientation as the electrostatic surface potential diagram in **e**. The possible entry point for electrons is indicated and the potential electron transferring residues are shown as sticks colored by atom type. **g** Potential electron wire through the protein core. The shortest distances between atoms in each residue are shown by black dashed lines. **h** Continuous wave X-band EPR spectrum (9.3 GHz, 160 K) with simulation (red) of *Td*AA15A in sodium phosphate buffer pH 7 and 10% v/v glycerol. Simulations were obtained with 15% of species 1, see Supplementary Table 3 for more details

content. Both species fall into a Peisach–Blumberg Type 2 classification, typical of LPMOs, although the somewhat reduced $|A_z|$ value for species 2, along with rhombic $g_x$ and $g_y$ values, shows some distortion away from axial symmetry, possibly influenced by the presence of the Ala89 side chain near the copper site and a degree of $d(z^2)$ mixing into the $d(x^2−y^2)$ SOMO. The speciation behavior was further investigated via EPR pH titrations in the presence and absence of 10% glycerol (Supplementary Fig. 11). These titrations confirmed the presence of two distinct copper active site coordination geometries, the relative ratio of which was

dependent on the pH and the exogenous ligand coordinating to the surface exposed active site.

**Characterization of a chitin-specific LPMO from *Thermobia*.** Most insects have fewer than five AA15 coding sequences per genome, while *Thermobia* and *Lepisma* (common silverfish) have expanded their repertoire to over 20 different isoforms (Supplementary Notes 2, 3), typically carrying an axial tyrosine, reminiscent of cellulose-active AA9s[1]. Only three of *Thermobia*'s LPMOs feature an axial phenylalanine (Supplementary Fig. 3), as seen in most chitin-active AA10s[2, 18, 25]. Phylogeny shows that

one of these sequences (contig GASN01010363.1, henceforth named *Td*AA15B) is located at the boundary between the two clades and might represent the evolutionary link between the two LPMO groups in this primitive insect (Fig. 5a). We produced the recombinant version of *Td*AA15B (Supplementary Fig. 12a, b) and carried out activity assays with the purified protein. MALDI-TOF MS analysis of the reaction supernatant revealed gallic acid-dependent formation of C1-oxidation products from both highly crystalline (α) and partially amorphous (β) chitin (Fig. 5b, c; Supplementary Fig. 12c–f), but not from cellulose (PASC and Avicel). MALDI-TOF MS analysis of crude extract from activity assays carried out with *Td*AA15B in presence of 10 mM EDTA failed to detect the release of both native and oxidized oligo-saccharides from both α- and β-chitin (Supplementary Fig. 13a, b), indicating that copper is crucial in activating the enzyme (as previously observed for *Td*AA15A).

**Possible role of AA15 LPMOs in chitin remodeling.** After cellulose, chitin is the second most abundant organic compound in nature[26] and constitutes the load bearing scaffold of both the exoskeleton and internal structures of arthropods, including the lining of the midgut (peritrophic matrix) and the tracheal respiratory system. We carried out shotgun proteomics on chitin-containing organs isolated from 3rd instar larvae of the model insect *Drosophila melanogaster* and found that *Dm*AA15B (corresponding to the annotated gene *CG4362*, Supplementary Note 4) represents roughly 3% of all CAZymes in the midgut (Fig. 5d). *Dm*AA15A (coded by gene *CG42749*, Supplementary Note 4), while being absent from the gut, makes up a notable 8% of the tracheal CAZymes, surpassed only by chitinases (GH18 family) (Fig. 5e). These proteomics data are confirmed by gene expression and in situ hybridization profiles collected from public databases (FlyAtlas, FlyBase, BDGP), showing that *DmAA15A* and *DmAA15B* are the most highly expressed LPMO genes during development and metamorphosis in *Drosophila*, and are specific for the trachea and midgut, respectively[27–29]. What is more, we found that 26% of all genes co-expressed with *Drosophila*'s LPMOs are involved in chitin metabolism (Supplementary Table 4) including *obst-A*, *obst-B*, *kkv*, *Edg78E*, *reb*, *Cht5*, *Cht6*, *Cht7*, *knk*, *pio*, *Cda4*, and *TwdlE*, which have demonstrated roles in chitin synthesis, deposition, and remodeling[30–39].

## Discussion

We present the characterization of a CAZy family of LPMOs (AA15) with putative roles in animal development and food digestion. The wide distribution of this LPMO family not only among complex animals but also in more primitive Eukarya, including oomycetes, protists and algae, indicates an ancient pre-Cambrian origin possibly dating back to the first build-up of atmospheric oxygen roughly 2 billion years ago. We propose that early arthropods first recruited AA15 LPMOs for endogenous chitin remodeling within the respiratory and digestive systems, and that this vital function is retained in modern insects. The important physiological role of the AA15 LPMOs in insects is confirmed by the effects induced by gene suppression. RNAi silencing of AA15 sequences in *Drosophila* leads to a range of deleterious effects including tracheal liquid clearance defects (*DmAA15A*)[40], death or significant adult morphology defects (*DmAA15B*)[40], and high lethality during pupation (*DmAA15C*, corresponding to gene *CG4367*)[41]. Similarly, RNAi gene knock-down of AA15 genes in *Tribolium castaneum*, a major pest of stored grain, affects metamorphosis and causes high pupal lethality (genes *TC016344*, *TC016345*, *TC016346*, *TC016347*, *TC016348*, *TC016349*, *TC016350*, *TC002263*, *TC015490*; iBeetle

website, http://ibeetle-base.uni-goettingen.de/) (Supplementary Table 5, Supplementary Note 5).

Our work indicates that the ancient insect order *Zygentoma*, including *Thermobia* and common silverfish, has co-opted endogenous AA15 LPMOs to boost asymbiotic cellulose digestion. Fossil records and phylogenomic analysis show that *Zygentoma* was one of the very first insect groups to colonize land, more than 400 Mya, and appear not to have evolved significantly since this ancient origin[8]. By revealing the abundance of endogenous lignocellulolytic enzymes in the gut of *T. domestica*, our proteomic studies strongly suggest that plant cell wall digestion by endogenous enzymes could be an ancestral trait in insects. In fact, endogenous cell-wall degrading enzymes (cellulases, β-glucosidases, β-1,3-glucanases, pectinases) have been reported in all major insect lineages, suggesting that the ancestral mechanisms for plant cell wall digestion in invertebrates were independent from microbial symbioses[42, 43]. The possession of these enzymes might help explain why insects thrived during the Carboniferous period (360–286 Mya), when plants fully colonized the land and atmospheric oxygen levels reached a peak of 35% compared to the current 21%. Such conditions would have likely favored the recruitment and expansion of carbohydrate-active oxidative enzymes for the degradation of abundant biomass. Our in depth phylogenetic, structural, and biochemical characterization strongly suggest that AA15 LPMOs are a key part of this ancestral mechanism and help explain why *T. domestica* is one of the most efficient cellulose degraders in the animal world.

We anticipate that the activity of AA15 LPMOs on industrially relevant polysaccharides, and their unprecedented distribution among diverse organisms, including important pest species and disease vectors, will open wide-ranging new areas for exploration.

## Methods

**Reagents.** 2,5-Dihydroxy benzoic acid, ascorbic acid, gallic acid, pyrogallol, hydroquinone, cysteine, quinic acid, p-coumaric acid, ferulic acid, $CuSO_4$, Trizma-Base (TRIS), 2-(*N*-morpholino)ethanesulfonic acid (MES), (4-(2-hydroxyelthyl)-1-piperazineethanesulfonic acid (HEPES), NaOH, 37% HCl solution, $Na_2HPO_4$, $NaH_2PO_4$, buffer standard solution pH 4 (phthalate), buffer standard solution pH 7 (phosphate), buffer standard solution pH 10 (borate), potassium ferricyanide, sucrose, glucose, ampicillin, and chloramphenicol were purchased from Sigma or Fisher Chemicals.

Avicel® PH-101 (microcrystalline cellulose) was purchased from Sigma and prepared by sonicating a suspension in 1.8 mM acetic acid with a Misonix sonicator, until particles were reduced to a size comparable to the one found in the crop of *Thermobia* fed on Avicel. The substrate was then washed several times in pure water until the pH reached 5.

Phosphoric acid swollen cellulose (PASC) was prepared as follows: 5 g of Avicel was moistened with water and treated with 150 mL ice cold 85% phosphoric acid, stirred on an ice bath for 1 h. Then 500 mL cold acetone was added while stirring. The swollen cellulose was filtered on a glass-filter funnel and washed three times with 100 mL ice cold acetone and subsequently twice with 500 mL water. PASC was then suspended in 500 mL water and blended to homogeneity.

Pure squid pen chitin (β-chitin) was kindly donated by Dominique Gillet (MAHTANI CHITOSAN Pvt. Ltd., India). Shrimp chitin (α-chitin) was purchased from Sigma and prepared by sonicating a suspension in 1.8 mM acetic acid with a Misonix sonicator. The substrate was then washed several times in pure water until the pH reached 5.

High purity pachyman, tamarind xyloglucan, barley β-glucan, lichenan (from Icelandic moss), mannan (borohydride reduced), pachyman, konjac glucomannan, carob galactomannan, larch arabinogalactan, and wheat arabinoxylan were purchased from Megazyme. Locust bean gum, carboxymethyl-cellulose (CMC), and beechwood xylan were purchased from Sigma.

**Rearing of *T. domestica* and *D. melanogaster*.** *T. domestica* live specimens were obtained from an online supplier and grown at 38 °C in plastic containers with holes on the lid for aeration. A small glass beaker with water was placed in each container to provide the appropriate moisture. Minerals were provided in the form of a multivitamin powder, proteins in the form of soy protein isolate. The carbon sources were powdered wheat straw, Whatman filter paper 1, Avicel, or blended oats. After feeding for at least 2 weeks on these diets, the animals were euthanized in ice and dissected under a stereo-microscope with sterile tools.

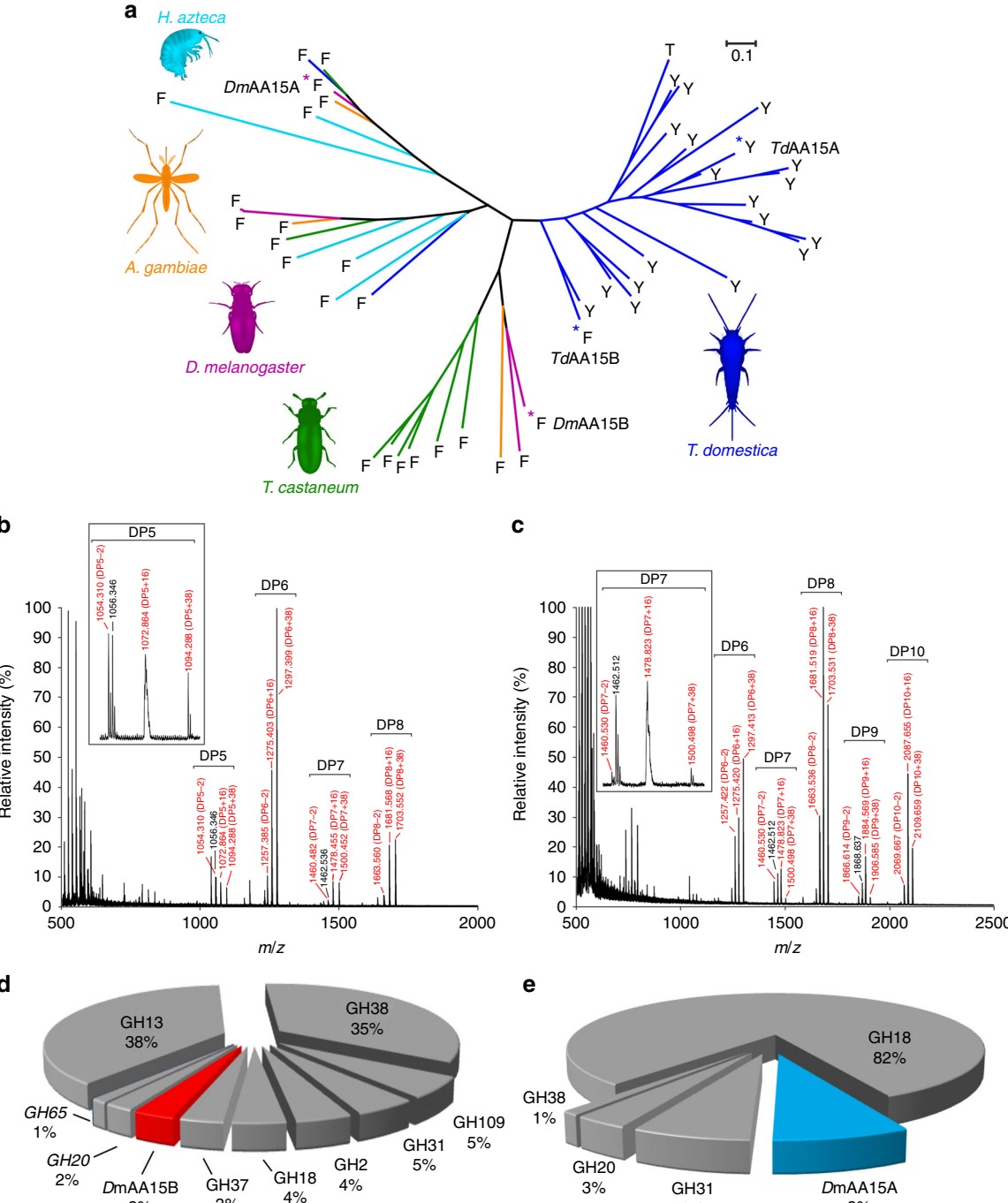

**Fig. 5** Phylogeny, biochemical characterization and tissue localization of insect AA15 LPMOs. **a** Maximum likelihood phylogenetic tree showing the AA15 coding sequences identified in the genome of three model insect species (*D. melanogaster*, *T. castaneum*, and *A. gambiae*), one model crustacean (*Hyalella azteca*), and in the transcriptome of *T. domestica* (Supplementary Notes 2, 4–7). Each tree branch is colored according to the species. The axial residue of each protein is indicated with a letter (Y = tyrosine, F = phenylalanine, T = threonine). Blue asterisks mark *Td*AA15A and the *Td*AA15B. Magenta asterisks mark *Dm*AA15A and *Dm*AA15B. MALDI-TOF MS spectra of products obtained after incubation of 4 mg mL$^{-1}$ α-chitin (**b**) and β-chitin (**c**) with 2 μM *Td*AA15B and 4 mM gallic acid for 24 h, showing native and oxidized oligosaccharides. The main peaks correspond to mono- or di-sodiated adducts of C1-aldonic acids, imparting +16 or +38 m/z respectively, relative to the mono-sodiated unoxidized form. Smaller peaks for the mono-sodiated lactone (−2) were also identified. In **b** and **c**, 100% relative intensity represents 2.9 × 10$^4$ and 1.3 × 10$^4$ arbitrary units (a.u.), respectively. Insets are expanded mass spectra for DP5 (**b**) and DP7 (**c**) products. Spectra of the negative control reactions are included in Supplementary Fig. 12c–f and do not show any native or oxidized species. **d** CAZymes identified in the proteome of the midgut from *D. melanogaster* larvae (whole tissue). The only LPMO identified in this sample is *Dm*AA15B. **e** CAZymes identified in the proteome of the tracheal system from *D. melanogaster* larvae (whole tissue). Only one LPMO (*Dm*AA15A) was detected and represents about 8% of the total CAZymes in the trachea. The most abundant GH family (GH18) includes several chitinases (accessions: X2JEB6, M9PGH3, Q9VFR3, A0A0B4LFJ1, D4G7B1, X2JA18, Q8MM24, M9NDS9), compatible with roles in chitin remodeling

*Drosophila* larvae were raised on a standard yeast, sugar, and agar medium at 25 °C. A standard wildtype stock, Canton-S, was used for dissections and analysis. Dissections were carried out at 3rd instar wandering larval stage.

**Shotgun proteomics.** Protein samples of *T. domestica* were prepared as follows. Crops from eight adults grown on a specific diet (wheat straw, filter paper, Avicel, or oats) were dissected in 50 mM sodium phosphate buffer pH 7 and the content (food particles and enzymes) was collected, added with 1% SDS, 1% beta-mercapto ethanol, 1% DTT, boiled for 10 min, centrifuged, and the supernatant shortly run in a 10% polyacrylamide gel.

Protein samples of *D. melanogaster* (wild type) were prepared as follows. Whole gut and tracheal system were dissected from 3rd instar larvae in phosphate buffered saline (137 mM NaCl, 2.7 mM KCl, 10 mM $Na_2HPO_4$, 1.8 mM $KH_2PO_4$, pH 7.4), added with 1% SDS, 1% beta-mercapto ethanol, 1% DTT, homogenized with a small plastic pestle, boiled for 30 min, and the supernatant shortly run in a 10% polyacrylamide gel.

In-gel tryptic digestion of proteins from both *Thermobia* and *Drosophila* was performed post reduction with DTE and S-carbamidomethylation with iodoacetamide. For *Thermobia*, resulting peptides were analyzed by label free LC-MS/MS over a 125-min gradient using a Waters nanoAcquity UPLC interfaced to a Bruker maXis HD mass spectrometer[44].

*Drosophila* digests were analyzed by LC-MS/MS using an UltiMate 3000 RSLCnano HPLC system interfaced with an Orbitrap Fusion hybrid mass spectrometer (Thermo). Peptides were eluted from a PepMap, 2 μm, 100 Å, C18 EasyNano nanocapillary column (75 μm x 150 mm, Thermo) at 300 nL min$^{-1}$ using gradient elution of two solvents: solvent A, aqueous 1% (v:v) formic acid; solvent B, aqueous 80% (v:v) acetonitrile containing 1% (v:v) formic acid (3–10% B over 8 mins, 10–35% B over 125 mins, 35–65% B over 50 mins). Positive ESI-MS and MS2 spectra were acquired using Xcalibur software (version 4.0, Thermo). Data-dependent acquisition was performed in top speed mode with a 1$^{-s}$ cycle. MS2 spectra were acquired in the linear ion trap with HCD activation energy of 32%.

Protein identification was performed by searching tandem mass spectra against a downloaded copy of the transcriptome of *T. domestica* (BioSample: SAMN02047119; Sample name: INSbttTSRAAPEI-29 - Thermobia domestica; SRA: SRS462938) and reference proteome of *D. melanogaster* (proteome ID: UP000000803) using the Mascot search program (version 2.5). Matches were passed through Mascot percolator to achieve a false discovery rate of <1% and further filtered to accept only peptides with expect scores of 0.05 or better. When tandem mass spectra were searched against the whole transcriptome of *Thermobia* (which included the full protein sequences before any downstream processing, such as signal peptide removal), we did not detect any masses compatible with the presence of the predicted N-terminal signal peptide for any of the LPMOs. We also set up a specific search in Mascot using histidine methylation as a variable parameter and interrogated the pool of mature LPMO proteins (without N-terminal peptide), which detected only masses compatible with a non-methylated histidine at the N-terminus. These results indicate that (1) mature LPMOs lack the N-terminal peptide; (2) mature LPMOs have a non-methylated N-terminal histidine.

Molar percentages of identified proteins were calculated from Mascot emPAI values by expressing individual values as a percentage of the sum of all emPAI values in the sample[45]. Proteins identified in the proteomics analysis were annotated via Blastx versus non-redundant NCBI databases. CAZy annotation was carried out using the CAZymes Analysis Toolkit (CAT) on the BioEnergy Science Center website (http://mothra.ornl.gov/cgi-bin/cat/cat.cgi) and dbCAN (http://csbl.bmb.uga.edu/dbCAN). Putative N-terminal signal peptide cleavage sites were predicted using the online tool SignalP 4.1 (http://www.cbs.dtu.dk/services/SignalP/).

All proteomic data sets, including raw data files, processed peak lists, and database search results are available to download from MassIVE (accessions MSV000081912 and MSV000081913) and ProteomeXchange (accessions PXD008657 and PXD008658). Deposited search results (.DAT and.mzIdentML) were generated using the latest version of Mascot (2.6).

**Phylogeny and classification of the AA15 LPMOs.** Phylogeny of the AA15 LPMO family as a whole was carried out as follows. The LPMO protein sequences identified in the transcriptome of *Thermobia* were searched via BlastP against NCBI non-redundant databases. A total of 192 AA15 sequences (169 sequences from curated NCBI genomes plus 23 *Thermobia* sequences) were analyzed for phylogeny. To avoid interference from the presence or absence of additional modules, the signal peptides and C-terminal extensions were removed. The resulting amino acid sequences corresponding to the catalytic domain were aligned using Muscle[46], operating with default parameters. A distance matrix was derived from the alignment using Blosum62 substitution parameters[47] and subsequently used to build a phylogenetic tree using the neighbor-joining method[48]. The resulting tree was visualized using Dendroscope[49] and edited with the graphic tools Gimp and CorelDraw.

AA15 protein sequences from *T. domestica*, *D. melanogaster*, *T. castaneum*, *A. gambiae*, and *H. azteca* were aligned using TCoffee[50] and a phylogenetic tree was build using the maximum likelihood method in Mega7[51].

**De novo transcriptome assembly of *Lepisma*.** Sequence read archive files for *Lepisma* sp. HW-2014 (accessions: SRR1184214 and SRR1184262) were downloaded from NCBI, the de novo transcriptome was assembled with Trinity[52] and interrogated via Blast search for the presence of orthologues of *Thermobia*'s AA15 sequences.

**Cloning the full-length gene of LPMO *GASN01030700.1*.** Genomic DNA was extracted from the legs of ten adult *Thermobia* specimens using the DNeasy Blood and Tissue Kit (Qiagen). External primers designed for contig *GASN01030700.1* were used to amplify the full gene (from start to stop codon) using genomic DNA as template and CloneAmp polymerase (Clontech) via nested PCR. The product, with estimated size of 4.5 kbp, was cloned with the StrataClone Blunt PCR Cloning Kit (Stratagene). The gene structure was then determined through Sanger sequencing using internal primers. Intron/exon boundaries were identified by comparing the full gene sequence with the coding sequence from the cDNA.

**RT-PCR and RNA-seq analysis of LPMO sequences.** For RT-PCR, salivary glands, crop, and anterior midgut were dissected from 10 *Thermobia* specimens grown on Avicel and the total RNA was extracted with the TRIzol® Reagent (Thermo Fisher Scientific). cDNA was generated with an oligodT primer using SuperScript® II reverse transcriptase (Thermo Fisher Scientific) in 20 μL reactions containing 300 ng RNA. 0.3 μL of cDNA was used as template in 15 μL PCR reactions to amplify LPMO coding sequences using Phusion® High-Fidelity DNA Polymerase (New England Biolabs) and sequence-specific oligonucleotide primers (Supplementary Table 6).

Gene expression levels for all putative LPMO sequences were determined by RNA-seq analysis. Raw reads were retrieved from NCBI (accession: SRR921648) and mapped onto the published transcriptome of *Thermobia* to determine normalized expression values (TPM = transcripts per kilobase million) using Salmon (part of the Galaxy toolshed)[53].

**Production of recombinant *TdAA15A* and *TdAA15B*.** The native sequences coding for *TdAA15A* and *TdAA15B* were cloned using cDNA generated from RNA extracted from *Thermobia*. Briefly, total RNA was extracted from one animal using the TRIzol® Reagent (Thermo Fisher Scientific) and cDNA was generated with an oligodT primer using SuperScript® II reverse transcriptase (Thermo Fisher Scientific).

The coding sequences starting from the codon of the catalytic histidine were amplified with oligonucleotide primers using Phusion DNA Polymerase (Thermo Fisher Scientific). A C-terminal Strep-tag® II (WSHPQFEK) was added to the C-terminus by PCR, and the amplicon was cloned into pET26b after the pelB leader sequence using the InFusion® HD Cloning Kit (Clontech).

The expression plasmid carrying the LPMO sequences was transformed into *E. coli* Rosetta 2 (DE3) pLysS (Novagen) via heat shock. A single colony was inoculated into LB medium plus 100 μg mL$^{-1}$ ampicillin and 34 μg mL$^{-1}$ chloramphenicol and grown overnight at 100 rpm at 30 °C; 10 mL of this starter culture were used to inoculate 1 L of M9 minimal salts medium containing 1% (w/v) glucose and the appropriate antibiotics. The cell culture was grown at 210 rpm at 37 °C until OD600 reached 0.7, then induced with 1 mM IPTG and left overnight at 20 °C. After protein expression, the cells were harvested, re-suspended in ice cold 50 mM Tris HCl pH 8 with 20% (w/v) sucrose, and left in ice for 30 min before centrifugation. The supernatant was discarded and the pellet was re-suspended in ice cold 5 mM $MgSO_4$ plus 100 μM AEBSF protease inhibitor and left in ice for 30 min. After centrifugation, the supernatant was collected, filtered, and the pH adjusted to 7.6 with 50 mM Na phosphate buffer. The periplasmic extract was then injected into a 5-mL StrepTrap HP column (Ge Healthcare) and, after washing with binding buffer, the protein was eluted with 2.5 mM desthiobiotin. Protein concentration was determined either with Bradford assay or from absorbance at 280 nm with a NanoDrop spectrophotometer (using molecular weight and extinction coefficient for the mature, strep-tagged protein). Five-fold excess copper was added as $CuSO_4$, then unbound copper and desthiobiotin were removed by passing the protein in a HiLoad$^{TM}$ 16/60 Superdex 75 gel filtration column (Ge Healthcare) equilibrated with 10 mM sodium phosphate buffer pH 7. The protein was then concentrated using Microsep$^{TM}$ Advance Centrifugal Devices (Pall Corporation).

**Thermal shift assay (Thermofluor).** The Thermofluor assay was conducted on the purified proteins with SYPRO® Orange Protein Gel Stain (Life Technologies) using an Mx3005P qPCR System (Agilent Technologies). The intensity of the fluorescence was measured at a temperature gradient of 25–95 °C and converted into a melting curve (fluorescence changes against temperature) to determine the melting temperature ($T_m$).

**ICP-MS.** Three technical replicates of reconstituted *TdAA15A* (after copper loading and gel filtration) were transferred to a digestion vessel. The samples were digested using 1 mL of nitric acid (70%, trace metal grade). The quartz vials were transferred to a microwave digestion system (Ethos Up) and heated to 200 °C (sealed vessel) for 15 min, then allowed to cool to room temperature. Once cooled, the reaction mixture was transferred to a 10-mL volumetric flask and diluted to

volume using deionised water (18 MΩ). Copper concentration was measured using an Agilent 7700x inductively coupled plasma-mass spectrometer (ICP-MS). Seven calibration standards were prepared using certified reference standards (multi-element environmental calibration standard, Agilent, Part number 5183–4688). Calibration curves had $r_2$ values of 0.998 or better. De-ionized water (18 MΩ) was used as a blank. The values obtained for the three protein samples were corrected against three technical replicates of the negative control (buffer only).

**In vitro activity assays.** Activity of the crop extract on a panel of substrates was determined by reducing sugar assay. Briefly, crops were dissected in 20 mM sodium phosphate buffer pH 6 containing 100 μM AEBSF (protease inhibitor) and the content fully re-suspended by pipetting. After centrifugation, the soluble portion (supernatant) was filtered through 0.22 μm porous membranes, quantified with the Bradford reagent and used for assays. Briefly, the typical 50 μL reaction was carried out in 96-well plates in 50 mM sodium phosphate buffer pH 6 with 2.8 μg of protein and 2 mg mL$^{-1}$ substrate. All reactions, including controls, were performed in triplicate. The microplate was incubated at 28 °C shaking at 320 rpm for 3 h, then 100 μL of DNS reagent were added to each reaction before heating at 100 °C for 5 min. Absorbance at 540 nm was measured with a micro-plate reader and nanomoles of reducing sugars released were determined based on absorbance obtained with glucose standards. The DNS reagent was prepared by mixing 0.75 g of dinitrosalycilic acid, 1.4 g NaOH, 21.6 g sodium potassium tartrate tetrahydrate, 0.53 mL phenol, and 0.59 g sodium metabisulfite in 100 mL pure water.

Plate assays were carried out by spotting 10 μL of soluble crop extract (concentration 0.56 mg mL$^{-1}$) on 1.2% agar plates containing 0.1% (w/v) substrate. After incubation at 28 °C for 16 h, the plates were covered with Congo Red solution (0.1 % w/v Congo Red in 5 mM NaOH) for 30 min at room temperature, then washed with 1 M NaCl and visualized. Activity was indicated by clearance zones. A 1/100 dilution of Celluclast® (Novozymes) was used as a positive control.

Typical reactions for LPMO characterization were carried out by mixing 1–4 mg mL$^{-1}$ substrate (PASC, Avicel, α-chitin, β-chitin) with purified $Td$AA15A/B (2 μM), 1–4 mM electron donor, in a total volume of 100 μL in 2 mL plastic reaction tubes. All reactions analyzed via MALDI were carried out in 50 mM ammonium acetate buffer pH 6 and incubated at 28 °C shaking at 600 rpm and the supernatant used for analysis.

Reactions used for product quantification and boosting experiments with $Td$AA15A were typically carried out in 50 mM sodium phosphate buffer pH 6 in triplicates of 100 μL each for 3 h at 600 rpm at 28 °C. Each reaction contained 2 μM purified $Td$AA15A, 1–4 mg mL$^{-1}$ substrate, and 1 mM electron donor. Commercial GH6 (cat. number E-CBHIIM, Megazyme), GH7 (cat. number E-CELTR, Megazyme), GH9 (cat. number CZ03921, ZNYTech), GH1 (cat. number E-BGOSAG, Megazyme), and GH18 (cat. number C6137-5UN, Sigma) were added to 100 μL reactions. After 3 h incubation, 400 μL of ethanol were added to stop the reaction, spun down and 400 μL of supernatant was transferred to new plastic tubes, dried down and re-suspended in 80 μL of pure water, filtered and analyzed via HPAEC.

**Product analysis by HPAEC.** Oligosaccharides were analyzed from undiluted samples via HPAEC using an ICS-3000 PAD system with an electrochemical gold electrode, a CarboPac PA20 3 × 150 mm analytical column and a CarboPac PA20 3 × 30 mm guard column (Dionex). Sample aliquots of 5 μL were injected and separated at a flow rate of 0.5 mL min$^{-1}$ at a constant temperature of 30 °C. After equilibration of the column with 50%–50% H$_2$O–0.2 M NaOH, a 30-min linear gradient was started from 0% to 20% with 0.5 M sodium acetate in 0.2 M NaOH and then kept constant for 20 min. The column was then washed with 0.2 M NaOH for 6 min and re-equilibrated for 4 min with 50%–50% H$_2$O–0.2 M NaOH before starting the next run (oligosaccharide method).

Glucose was analyzed with the following HPAEC program (monosaccharide method). After equilibration of the column with 100% H$_2$O, sample aliquots of 5 μL were injected and separated at a flow rate of 0.5 mL min$^{-1}$ at a constant temperature of 25 °C. The column was washed with 100% H$_2$O for 10 min, followed by 9 min of 99%–1% H$_2$O–0.2 M NaOH. The column was then washed with 0.2 M NaOH for 6 min and re-equilibrated with 100% H$_2$O before injection of the next sample.

Integrated peak areas were compared to mono and oligo-saccharide calibration standards (glucose, cellobiose, cellotriose, cellotetraose, cellopentaose, cellohexaose, N-acetylglucosamine, chitobiose, chitotriose, chitotetraose, chitopentaose) purchased from Megazyme.

**Product analysis by mass spectrometry.** One microliter of reaction supernatant was mixed with an equal volume of 20 mg mL$^{-1}$ 2,5-dihydroxybenzoic acid (DHB) in 50% acetonitrile, 0.1% TFA on a SCOUT-MTP 384 target plate (Bruker). The spotted samples were then dried down in a vacuum desiccator before being analyzed by mass spectrometry on an Ultraflex III matrix-assisted laser desorption ionization-time of flight/time of flight (MALDI/TOF-TOF) instrument (Bruker)[54].

Sample permethylation was carried out according to Ciucanu and Kerek[55]. Spotted samples were analyzed by MS using 2,5-DHB matrix with 0.1% TFA on an AB-Sciex 4700 (for MALDI-TOF) and Ultraflex III MALDI/TOF-TOF instrument (Bruker). Data were collected using a 2-kHz smartbeam-II laser and acquired on

reflector mode (mass range 300–3000 Da) for MS analysis and on LIFT-CID for MS/MS analysis using argon as collision gas. FlexControl and FlexAnalysis softwares were used for data acquisition and analysis. On average, about 10,000 shots were used to obtain high-enough resolution. MS/MS fragmentation patterns were named according to Domon and Costello[56].

**X-ray crystallography of $Td$AA15A.** Sitting-drop crystallization screens were set up using copper-loaded $Td$AA15A at 10 mg mL$^{-1}$ using fomulatrix NT8 robotics. Initial crystal hits were obtained in the JCSG Core I and II screens (Qiagen), conditions F11 and H11, respectively. These crystals were subsequently optimized in further sitting-drop vapor diffusion experiments mixing 0.2 μL of the protein at 10 mg mL$^{-1}$ with 0.1 μL of crystallization solution −0.1 M sodium citrate pH 5.5, 0.1 M LiCl, and 10–25% w/v polyethylene glycol 6000 (PEG-6000). All screens were performed at 20 °C.

Crystals were cryo-protected by soaking in mother liquor supplemented with 20% ethylene glycol before being plunged in liquid nitrogen. Data were then collected at the ESRF, MASIF-1 beamline at a fixed wavelength of 0.966 Å. Ten data sets were collected without manual intervention, five of which were collected using the MXPressE_SAD protocol to allow attempts at experimental phasing using the weak anomalous signal that would be obtained from the copper at this wavelength, and five data sets were collected using the MXPressE protocol to provide the best possible native data. All data sets were indexed using XDS[57]. Individual data sets were processed using CCP4[58] but these did not contain sufficient anomalous signal to allow structure determination. All five data sets collected using the MXPressE_SAD method were, therefore, combined and scaled using BLEND[59] to 2 Å resolution. The structure was then successfully determined from the copper anomalous signal using SHELX[60]. The initial structure was rebuilt using BUCCANEER[61] and this model was then refined against the best native data set at 1.1 Å resolution. Subsequent rounds of manual rebuilding and refinement were performed in COOT[62] and REFMAC5[63], respectively. The quality of the model was monitored throughout rebuilding and refinement using MolProbity[64], with the final model containing a single Ramachandran outlier (Ser33) and 96.9% of residues in the favored region of the Ramachandran plot. Data processing and structure refinement statistics are shown in Supplementary Table 2.

The structure and accompanying structure factors have been deposited in the Protein Data Bank with accession code 5MSZ.

**ConSurf analysis.** For ConSurf[21] analysis, we generated an alignment using 193 publicly available sequences defined as being in this LPMO family in CAZy, using MUSCLE[46]. The 21 sequences identified in the current study were then added to the alignment using MAFFT[65], giving a final alignment containing 214 sequences from the same family. This alignment was then uploaded to the ConSurf[21] server for analysis, ensuring that only LPMOs in the same family were analyzed. The ConSurf scores were visualized on the protein surface using PyMol.

**UV-vis spectroscopy.** The UV-vis spectrum of Cu(II)-$Td$AA15A was collected on a 1 mM sample of protein in 20 mM sodium phosphate buffer pH 7 using a Shimadzu UV-1800 spectrophotometer.

**Electron paramagnetic resonance spectroscopy.** Continuous wave X-band frozen solution EPR spectra of single samples of 0.2 mM solutions of Cu(II)-$Td$AA15A (in 10% v/v glycerol) at pH 7.0 (50 mM sodium phosphate buffer) and 160 K were acquired on a Bruker EMX spectrometer operating at ~9.30 GHz, with a modulation amplitude of 4 G and microwave power of 10.02 mW. EPR pH titrations were performed as a single experiment on a 0.3-mM sample of Cu(II)-$Td$AA15A in a mixed buffer composed by sodium acetate, MES, HEPES, and TRIS at the concentration of 10 mM for each component, with or without 10% v/v glycerol. The pH was adjusted using 1 M NaOH or 1 M HCl solutions and measured directly in the protein sample using an InLab® micro pH electrode from Mettler Toledo connected to a Radiometer Analytical ION450® pH-meter calibrated using standard buffer solutions at pH 4.01, 7.00, and 10.01. The EPR spectra were collected between pH 5.0 (±0.1) and pH 8.5 (±0.1) every 0.5 pH unit. Slight protein precipitation was visible at pH 5 (predicted pI of the protein is 4.9), but the process was completely reversible upon change of pH and the EPR spectra were not affected. To investigate redox properties, a 0.2-mM sample of $Td$AA15A in 20 mM sodium phosphate buffer pH 6 was incubated with 100-fold excess of sodium ascorbate, with EPR spectra collected before and after addition of the reducing agent. Gallic acid was added to the protein in the same conditions, but no decrease in the copper signal was visible over a 3-h period, which—since gallic acid is clearly active in the assays—implies that the reduction potential of the copper active site is affected by the presence of substrate. Similarly, 20 equivalents of potassium ferricyanide were added to a 0.2-mM sample of $Td$AA15A in 20 mM sodium phosphate pH 6 with 10% glycerol and EPR spectra recorded before and after addition. Spectral simulations were carried out using EasySpin 5.2.6[66] integrated into MATLAB R2016a software[67] on a desktop PC. Simulation parameters are given in Supplementary Table 3. $g_z$ and $|A_z|$ values were determined accurately from the three absorptions at low field. It was assumed that $g$ and $A$ tensors were axially coincident. Accurate determination of the $g_x$, $g_y$, $|A_x|$ and $|A_y|$ was not possible due to the presence of two species, although it was noted that satisfactory simulation

could only be achieved with the particular set of values reported in Supplementary Table 3. Furthermore, it was noted that the simulations were improved by the addition of coupled nitrogen atoms. For species 2, the exact value of the coupling could not be determined given the lack of well resolved superhyperfine (SHF) coupling, therefore only a range is reported. For species 1, the EPR spectra collected at pH 8.5 in the absence of glycerol (Supplementary Fig. 10) presented more resolved SHF coupling and the simulations confirmed the presence of three coordinated nitrogen atoms (Supplementary Table 3). Raw EPR data are available on request through Research Data York (DOI:10.15124/bd09e86b-9d92-4802-9337-18b138e7abb7).

**Data availability**. All proteomic data sets, including raw data files, processed peak lists, and database search results are available to download from MassIVE (accessions MSV000081912 and MSV000081913) and ProteomeXchange (accessions PXD008657 and PXD008658). Deposited search results (.DAT and.mzIdentML) were generated using Mascot v2.6. Atomic coordinates and structure factors for the X-ray structure of *TdAA15A* were deposited in the Protein Data Bank under accession code 5MSZ. Raw EPR and UV-vis data are available on request through Research Data York (doi.org/10.15124/bd09e86b-9d92-4802-9337-18b138e7abb7). All other data are available from the corresponding authors upon reasonable request.

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

## Acknowledgements

We thank the European Synchrotron Radiation Facility (ESRF), France, for synchrotron beam time and assistance. We also thank Bernhard Misof, Karen Meusemann, and the 1KITE Project for help and support, and Dominique Gillet for kindly providing β-chitin. This work is funded by the UK Biotechnology and Biological Sciences Research Council (grant number BB/L001926/1). The York Centre of Excellence in Mass Spectrometry was created thanks to a major capital investment through Science City York, supported by Yorkshire Forward with funds from the Northern Way Initiative, and subsequent support from EPSRC (EP/K039660/1; EP/M028127/1).

## Author contributions

F.S. carried out sample isolation for shotgun proteomics, analysis of proteomics data, RNA and DNA extractions, cloning, enzyme activity assays, HPAEC, and MS analysis of reaction products; K.B., L.D.G., and R.S. helped with HPAEC analysis; F.S., K.B., L.E., and G.P. carried out animal rearing and dissection; F.S. and L.E. performed heterologous expression and protein purification; G.R.H. crystallized the proteins, collected and analyzed the crystallographic data, solved the crystal structures and made structural figures and tables; P.H.W. and L.C. conceived the EPR study; L.C. carried out EPR experiments and simulations; B.H. and Y.L. performed bioinformatics analyses and alignments; P.D., T.T., and R. M. did the analysis of the permethylated reaction products; A.A.D. and R.B. analyzed protein samples via LC/MS-MS and ESI-FTICR-MS; F.S., G.R. H., S.T.S., P.H.W., G.J.D, N.C.B., and S.J.M. organized the data and wrote the manuscript; all authors reviewed and commented on the manuscript.
