## [Peer Review File · Nature Communications]

Reviewers' comments:

Reviewer #1 (Remarks to the Author):

The manuscript describes identification of LPMOs, one of which has very unique promiscuous specificity for cellulose and chitin, from firebrat's gut. This work will establish a novel LPPO family (AA14), which firstly includes animal enzymes. This is an excellent paper providing a novel insight into the evolutionary aspect of chitin/cellulose-degrading enzymes in insects, and information for developing novel tools to combat harmful insects. The data are solid, and the paper is clearly written.

Major points:

1. Definition of a new CAZy family. As to the sequence similarity to known LPMO families, the descriptions are apparently inconsistent (Page 4): "sharing distant sequence similarity to .." "similarity to LPMO was most evident in a conserved N-terminal histidine brace .." / "the lack of significant similarity with previously established LPMO families ..". To avoid misunderstanding as if the authors (including the official CAZy team) intentionally divide/duplicate and make a new family, rough criteria must be indicated. In this case, e-values or some other quantitative indicator must be shown for the "significant" and "distant" similarities.
2. Indicate the characterized enzymes (TdAA14A, TdAA14B, Dm14AA14A, and DmAA14B) in Figures 2, 5, S3, and S4. In the legend of Supplementary Fig. 5a, describe that lane a is TdAA14A.
3. Did the authors check the LPMO activity of the EDTA-treated protein samples?
4. Space group of the crystal (Supplementary Table 1). As far as I know, the orthorhombic space group (P 2 21 21) must be reindexed to move the unique axis to c-axis (P 21 21 2).

Minor points:

1. Page 7, line 4 from the bottom: "..., and the recombinant protein was purified" (not sure)
2. Page 8, lines 8-9: "all three substrates" but only Avicel and chitin are shown in Fig. 3ab.
3. Page 8, line 2 from the bottom: "beta-glucosidase" (beta character)
4. Page 18, Fig. 4 legend: (b) What kind of electron density? 2mFo-mFc ?
5. Page 13, line 8 from the bottom: feature an axial phenylalanine (Supplementary Fig. 3).
6. Legends of Supplementary Figs. 2 and 3. Spell out "50" and "23" at the head of the sentences.
7. Supplementary Fig. 3: In the legend indicate that the sequences are mature ones without signal sequences.
8. Supplementary Fig. 5 legend: Strong and specific amplification (not sure)
9. Supplementary Table 1: No need to write the alpha/beta/gamma angles for orthorhombic space groups. B-factors should have square angstrom units.

Reviewer #2 (Remarks to the Author):

The manuscript by Sabbadin et al describes a beautiful, carefully conducted, and important study. The paper describes the characterization of insect LPMOs and demonstrates their (possible) biological roles. This is yet another exciting step in unravelling the impact of LPMOs in biology and industry. I am convinced that these enzymes, discovered in 2010, will not only change how we think about enzymatic biomass conversion, but also have major impact on other concepts in biology. The present paper provides a groundbreaking example.

My only main criticism is that the authors are overselling, which really is not necessary in this case. Even after toning down a little, this still is a great paper, of major importance for a wide

scientific audience, and with lots of novelty. So, while I am enthusiastic about this paper I would insist that some obvious mistakes related to this overselling are corrected. Notably, while some of my comments may seem harsh, I am really only asking for minor corrections. This is a great paper.

There are two elements of overselling:

(A) The "discovery route" towards finding the AA14s. Finding these enzymes could have been done by simple bioinformatics, as already pointed out in 2012 (see below). The "problem" is that CAZy, uses another sequence cut-off than Pfam, which, I would argue, is also widely used. I wish to stress that, nevertheless, the -omics data in this paper are important and exciting. I am only asking for some textual adjustments.

(B) It is difficult to avoid getting the impression that this paper also is about arthropod development and chitin remodeling. In my view it is not. However, combining the present data with literature data gives important novel insights into arthropod development, as pointed out in the Discussion section. In light of this, while the title is okay, some textual adjustments are needed elsewhere.

Suggestions for changes:

(1) The abstract should be toned down (a little), in light of my comments.

(2) The present paper shows, for the first time, that a putative insect LPMO is active. This is very important, especially in the light of the -omics data which show that LPMOs are highly expressed and in the light of the presence of similar LPMOs in other higher eukaryotes. However, the fact that putative LPMOs occur in e.g. insects is not new; this has been predicted as early as in 2012 in a review by Horn et al., 2012 by simply searching genomes with the Pfam HMM (PF03067) (such a search yields Drosophila AA10s). CAZy is fantastic, and it is appropriate to create the AA14 class, but the reader deserves to know a little more about why CAZy creates several classes within the same Pfam entry. Also, in light of the above, the fact that this LPMO founds a new class in CAZy is as such not a sign of great novelty (biochemically, the characterized enzyme does not show great novelty neither). I suggest the following adjustments:

2a. Previous predictions concerning the presence of LPMOs in insects and other higher eukaryotes should be mentioned and citations should be added.

2b. The rationale behind creating the AA14 class, and the relation between CAZy and Pfam needs to be explained better, in light of the comments above.

2c. The authors should go through their manuscript and carefully consider the use of terms such as "new", "unique", in light of the above comments. Some specific suggestions:

- Lines 26-28 needs rephrasing (this sentence is suboptimal in several ways, including an overstatement of novelty; see also comment #3)

- Lines 51-52 are a little ambiguous

- Line 60: this family is uncharacterized in the CAZy world, which I happen to like a lot, but not in the Pfam world (where AA10 and at least some of the AA14s are in the same family, for which both chitin activity and cellulose activity have been shown, in 2010 and 2011, respectively).

- Lines 85-90; I am quite sure that a simple Pfam search would have told you that there were putative LPMOs here.

- Lines 93-95; here the use of the word "unexpectedly" is unacceptable.

- Lines 100-102; significance comes in many forms....

- Line 198: the structure does not unravel features related to promiscuity (see other comment), and this promiscuity is not "unusual". One of the first cellulose-active AA10s (PF03067) that was ever structurally characterized also shows this promiscuity (Forsberg et al., 2014; PNAS, ScLPMO10B)

- Line 206-207: here you go..... (By the way, it would be appropriate to cite the CBP21 structure paper, which dates back to 2005 and presents the first structure of an LPMO)

- Line 227: "entirely new". Really? LPMO surfaces vary a lot.

- Lines 314-315
- Line 339: delete "uniquely"
- Line 432: delete "new"

(3) The proteomics work is important and interesting, but I think it is wrong to say (or suggest) that this part of the study was essential for finding an active LPMO from an insect. Therefore, I suggest that the discovery aspect of the proteomics part of the manuscript is toned down a little. This is another reason to rephrase lines 26-28 (Abstract). Please make necessary minor adjustments throughout the manuscript.

(4) In my view, as outlined above, there are no experiments in this study on arthropod development and "chitin remodeling". Therefore, I suggest some rephrasing:

- Lines 64-66
- The section starting on page 259 should have another title, and/or perhaps be split in two. The first two paragraphs of this section have nothing to do with deciphering the true biological function of the LPMOs. This is different for the third paragraph (and part of the Discussion section). I find lines 274 - 278 questionable; this is overselling as I see it. (Nb. Line 273: what is the basis for the statement that termites "have not diversified their LPMOs" - How can we say that while we do not know the structural determinants of substrate specificity?)

(5) All MS pictures should be carefully checked for errors in peak labeling (e.g. Fig. 5c - incomplete labeling) and peak labeling should be made more clear (using helper lines?). Perhaps use an insert with zoom-in? Near line 165, the paper should contain an explicit statement concerning the possible occurrence of C4 oxidation. Could the native peaks visible in Fig. 3A be due to a C1/C4 mixed activity? Or could they be due to on-column degradation of C4 products, as pointed out by Westereng et al., 2016? I am a little concerned when it comes to the MS spectra in Fig. 3 because the relative spacing between the -2, +16 and +38 products is not the same in panels A and B. I do not understand how that is possible.

(6) Several Figures show synergy experiments based on single time point measurements. This is fine for the purpose of this study, but all quantification of these synergies must be deleted from the manuscript (including suppl material). LPMO kinetics are notoriously non-linear and many factors come into play that may not even relate to LPMO activity as such. True quantitative impressions of synergy may only be obtained by studying progress curves.

Other issues:

1. Lines 30-33 need some rephrasing. I do not really see how the "In-depth characterization" warrants the speculations in the second half of the sentence. Both are worth mentioning, but perhaps without the connection.
2. Line 113 & 124. It is extremely interesting that 21 LPMOs were detected, i.e. a very high fraction of the total number. This is not usually the case in proteomics studies of fungi, where a lesser fraction of the total numbers of LPMO tends to be detected. Does this really mean that a large number of the LPMOs is clearly expressed in *T. domestica*? Or is this due to e.g. another cut-off value? Perhaps some sort of quantitative statement could be made? Are all 21 clearly there? In light of this, it would also be interesting to know a little more about the RT-PCR experiments. How many of the 23 LPMOs were tested by RT-PCR and what were the results? (Nb. I am not asking for more RT-PCR experiments, but if the authors have more information already, this should be included)
3. Line 120-122. How well was methylation assessed (for example, for how many of the LPMOs

was the N-terminal peptide actually detected in the proteomics study?)

4. Line 210 and below: It would be nice to explain to the reader what is meant/implied by "two species".

5. Line 224. It would be appropriate to cite Kracher et al., 2016.

6. 328-329: Another selling statement. How could this work? Lines 329-337 could be deleted; this is speculation (how could this work?)

7. Line 469-472: Any reason for not using gene synthesis?

8. Line 473-475: Unclear; add more details.

9. Line 502 and onwards: Where no protease inhibitors used?

10. Fig. S3: Please indicate the protrusion.

11. Legend to Fig. S5: which LPMOs were tested by RT-PCR? Does the list include TdAA14A? Please indicate.

12. Legend to Fig. 4c: Are these four LPMOs C1, C4 or C1&C4 oxidizers? I would suggest citing the papers behind each of these structures.

Reviewer #3 (Remarks to the Author):

This paper describes the cloning, expression, protein purification, and kinetic, structural, and EPR characterizations of a recombinant lytic polysaccharide monooxygenases (LPMO) present in the digestive proteome of *Thermobia domestica*. There are several examples of LPMOs reported in the literature, the novelty of the paper is that the protein characterized represents the first LPMO from an animal genome.

I have some concerns with the MS that should be addressed before publication in Nature comms.

1) On pages 7-9. Biochemical characterization. The role of copper in the catalytic mechanism should be better explained. Copper is incorporated into the apoprotein on addition of copper excess. Apparently, the apoprotein is not catalytically competent; however, it is not clear whether the catalytic activity of the apoprotein was checked or not.

Copper content and copper/protein ratio of reconstituted LPMO is not reported, which should be included in the MS.

The redox role of the copper center in catalysis, which is slightly mentioned in the MS, should be better explained. Authors provide experimental evidences highlighting the structural role played by copper but only a brief mention on the copper redox role is given. Although a complete redox characterization should be desirable, the author should at least provide EPR information of the copper centers by reporting the copper behavior towards conventional reductants such as ascorbate and dithionite and oxidants such as ferricyanide. This standard procedure should give at least information of the order of magnitude of the copper reduction potential and perhaps helps to discriminate the EPR signals associated with the two copper species that supposedly are present in LPMO (see below also). The EPR behavior towards the electron donors used in the catalytic assays should be included and analyzed. The UV-vis properties, mainly those associated with the copper

centers, should also be reported.

2) On page 10. Structural and spectroscopic characterization.

The EPR spectrum of LPMO is interpreted on the basis of two superposed spectral components with different EPR parameters. Superhyperfine interactions with the coordinating ligands is also included, but one has to go the supplementary material to understand the latter.

The simulation assumptions are a matter of criticism. Firstly, one could suppose two spectral components to simulate the spectrum, but not to include N superhyperfine interactions, as no experimental evidences can be detected in the spectrum. Furthermore, isotropic N-couplings (Table S2) do not have physical sense. The assumption of two spectral components from the single EPR spectrum reported is rather controversial, as additional EPR experiments should be necessary to obtain such a conclusion. There are several possible explanations for the EPR spectrum of LPMO. Note that the X-ray crystal data, which are taken with 1.1 Å resolution, show only a single metal center with no evidences of disorder, in which case the simplest interpretation for the EPR experiment in solution is some metal site inhomogeneity. I think that the number of such conformations cannot be assessed with the present data. In other words, the author's proposal would require additional conventional EPR experiments such as EPR redox titration complemented with spin quantifications, and more specialized ones such as EPR at different MW frequencies and N-hyperfine interaction detection by more sophisticated magnetic resonance techniques.

MS and supporting information. "Accurate determination of perpendicular values and superhyperfine coupling constants was not possible due to second order nature of the spectrum in this region" Please, clarify the meaning of "second order nature etc" Do authors want to mean second order corrections usually employed in perturbation theory?

Fig. 4, d. Simulation of each spectral component should be included.

On page 10, lines 2017-210. "While the third, non-coordinating active site residue of TdAA14A is a tyrosine (Tyr184) as in most AA9s, the positioning of Ala89 is reminiscent of AA10s (Fig. 4c) and such features are reflected in the rhombic EPR spectrum of the Cu(II) form of TdAA14A (Fig. 4d, Supplementary Table 2)"

It sounds strange that non-coordinating copper ligands determine the symmetry of the Cu(II) g-matrix. Note also the copper as depicted in Fig. is 3-coordinated. This is rather unusual for a Cu(II) ion, which deserves some comment.

On page 10, lines 212-213 and 216-219. "The mixed species observed and the variation in their ratio in response to exogenous ligands and pH reveal" These EPR data should be shown either in the MS or as supporting information.

Note also that glycerol was added to the EPR samples as cryoprotectant. One should avoid this very old procedure when possible, as glycerol is a potential Cu(II) ion ligand. EPR studies with and without glycerol should be analyzed and compared.

We thank the three reviewers for the positive evaluation and constructive comments about our manuscript. We believe that the additional experiments and revisions have addressed the raised issues, significantly improving both clarity and robustness of the presented work. Below are our responses to the points raised in the reviewers` report.

Note to the Editors and reviewers

We have been informed that a paper has been recently accepted (now under publication) describing a new family of LPMOs (unrelated to the one presented in our work), and that such family has been officially named AA14 and will soon be added to the CAZy database. The LPMO family discussed in our manuscript has therefore been re-named AA15.

Reviewer #1 (Remarks to the Author):

The reviewer stated: "This is an excellent paper providing a novel insight into the evolutionary aspect of chitin/cellulose-degrading enzymes in insects, and information for developing novel tools to combat harmful insects. The data are solid, and the paper is clearly written."

The reviewer asked us to address a number of issues, which are presented in italicised text below, along with our responses.

Major points:

1. Definition of a new CAZy family. As to the sequence similarity to known LPMO families, the descriptions are apparently inconsistent (Page 4): "sharing distant sequence similarity to .." "similarity to LPMO was most evident in a conserved N-terminal histidine brace .." / "the lack of significant similarity with previously established LPMO families ..". To avoid misunderstanding as if the authors (including the official CAZy team) intentionally divide/duplicate and make a new family, rough criteria must be indicated. In this case, e-values or some other quantitative indicator must be shown for the "significant" and "distant" similarities.

We have corrected and expanded the paragraph "Phylogeny and sequence analysis", describing the use of HMM models to rigorously define a new LPMO family. We have also included a supplementary table (Supplementary Data 5) showing the results of the HMMer search of 240 AA15 sequences against the HMMs of each LPMO family available in the CAZy database.

2. Indicate the characterized enzymes (*TdAA14A*, *TdAA14B*, *Dm14AA14A*, and *DmAA14B*) in Figures 2, 5, S3, and S4. In the legend of Supplementary Fig. 5a, describe that lane a is *TdAA14A*.

With the only exception of Supplementary Figure 4, where the lack of detail does not allow the clear separation of the single branches of the phylogenetic tree, we have amended all other figures and legends as suggested by the reviewer.

3. Did the authors check the LPMO activity of the EDTA-treated protein samples?

To address it, we have carried out *in vitro* activity assays with both *TdAA15A* and *TdAA15B* using the same experimental conditions shown in Fig. 3a-b and Fig. 5b-c, in presence of 10 mM EDTA. MALDI-TOF MS analysis of the crude supernatant did not reveal the formation of native nor oxidized oligosaccharides, indicating that the apo-enzymes were completely inactive. We have added the following sentence (paragraph “Biochemical characterization”): “MALDI-TOF MS analysis of crude extract from activity assays carried out with Cu-loaded *TdAA15A* in presence of 10 mM EDTA failed to detect the release of both native and oxidized oligosaccharides (Supplementary Fig. 7 a,b), indicating that the apo-enzyme is not active and that copper is essential for activity.” Similarly, we have now mentioned the result of the assay with *TdAA15B* in paragraph “Characterization of a chitin-specific LPMO from *Thermobia*”: “MALDI-TOF MS analysis of crude extract from activity assays carried out with *TdAA15B* in presence of 10 mM EDTA failed to detect the release of both native and oxidized oligosaccharides from both α and β -chitin (Supplementary Fig. 13 a,b), indicating that copper is crucial in activating the enzyme (as previously observed for *TdAA15A*).” The spectra from reactions of both enzymes with EDTA have been added as Supplementary Figures 7 and 13.

Supplementary Figure 7 | MALDI-TOF MS of activity assays with *TdAA15A* in presence of EDTA.

MALDI-TOF MS analysis of *in vitro* negative control activity assays with purified *TdAA15A*, under the same experimental conditions as in Figure 3a, b. The panels show spectra of products obtained after incubation of 4 mg mL⁻¹ microcrystalline cellulose (a) or β -chitin (c) with 2 μ M *TdAA15A*, 4 mM gallic acid and 10 mM

EDTA. The spectra do not show detectable amounts of native or oxidized cello-oligosaccharides. In **a** and **b**, 100% relative intensity represents 0.9×10^4 and 1.0×10^4 arbitrary units (a.u.), respectively.

Supplementary Figure 13 | MALDI-TOF MS of activity assays with *TdAA15B* in presence of EDTA.

MALDI-TOF MS analysis of *in vitro* negative control activity assays with purified *TdAA15B*, under the same experimental conditions as in Figure 5b, c. The panels show spectra of products obtained after incubation of 4 mg mL^{-1} α -chitin (**a**) and β -chitin (**b**) with $2 \text{ }\mu\text{M}$ *TdAA15B*, 4 mM gallic acid and 10 mM EDTA. The spectra do not show detectable amounts of native or oxidized cello-oligosaccharides. In **a** and **b**, 100% relative intensity represents 2.9×10^4 and 1.3×10^4 arbitrary units (a.u.), respectively.

4. *Space group of the crystal (Supplementary Table 1). As far as I know, the orthorhombic space group (P 2 21 21) must be reindexed to move the unique axis to c-axis (P 21 21 2).*

In the past, this was indeed the case, however, the pdb now accepts “non-standard” indexing and has done for some time. There are in fact more than 500 structures in the pdb that are in the space group P 2 21 21. Modern crystallographic software uses the convention that cell dimensions are assigned such that the length order is $a < b < c$ and hence the space group in this case was assigned as P 2 21 21. We would favour leaving this as it is at present as the structure has been deposited and this is now an accepted space group.

Minor points:

1. *Page 7, line 4 from the bottom: "., and the recombinant protein was purified" (not sure)*

The sentence was rephrased as follows: “The coding sequence representing one of the most abundant *T. domestica* LPMOs (contig GASN01405718.1, henceforth termed *TdAA15A*) was cloned and expressed in *Escherichia coli* with a C-terminal strep-tag. The recombinant protein was purified from the bacterial periplasm by affinity chromatography (Supplementary Fig. 5c) and characterized.”

2. *Page 8, lines 8-9: "all three substrates" but only Avicel and chitin are shown in Fig. 3ab.*

We apologize for the lack of clarity. We have moved the reference to PASC a few lines below, where the results of the permethylation experiment are presented.

3. *Page 8, line 2 from the bottom: "beta-glucosidase" (beta character)*

Duly noted and changed.

4. *Page 18, Fig. 4 legend: (b) What kind of electron density? 2mFo-mFc ?*

We have updated the figure legend.

5. *Page 13, line 8 from the bottom: feature an axial phenylalanine (Supplementary Fig. 3).*

We appreciate the suggestion and have changed the text accordingly.

6. *Legends of Supplementary Figs. 2 and 3. Spell out "50" and "23" at the head of the sentences.*

Both numbers have now been changed to text.

7. Supplementary Fig. 3: In the legend indicate that the sequences are mature ones without signal sequences.

We have included this detail in the legend title.

8. Supplementary Fig. 5 legend: Strong and specific amplification (not sure)

The sentence has been deleted.

9. Supplementary Table 1: No need to write the alpha/beta/gamma angles for orthorhombic space groups. B-factors should have square angstrom units.

We have maintained a row in the table to show the space group angles for clarity, but have only added 90.0° to indicate that all the angles are equal. Thanks also for pointing out our oversight on the B-factor units. The units have now been included in the table (now Supplementary Table 2).

Reviewer #2 (Remarks to the Author):

The manuscript by Sabbadin et al describes a beautiful, carefully conducted, and important study. The paper describes the characterization of insect LPMOs and demonstrates their (possible) biological roles. This is yet another exciting step in unravelling the impact of LPMOs in biology and industry. I am convinced that these enzymes, discovered in 2010, will not only change how we think about enzymatic biomass conversion, but also have major impact on other concepts in biology. The present paper provides a groundbreaking example.

My only main criticism is that the authors are overselling, which really is not necessary in this case. Even after toning down a little, this still is a great paper, of major importance for a wide scientific audience, and with lots of novelty. So, while I am enthusiastic about this paper I would insist that some obvious mistakes related to this overselling are corrected. Notably, while some of my comments may seem harsh, I am really only asking for minor corrections. This is a great paper.

There are two elements of overselling:

(A) The “discovery route” towards finding the AA14s. Finding these enzymes could have been done by simple bioinformatics, as already pointed out in 2012 (see below). The “problem” is that CAZy, uses another sequence cut-off than Pfam, which, I would argue, is

also widely used. I wish to stress that, nevertheless, the –omics data in this paper are important and exciting. I am only asking for some textual adjustments.

(B) It is difficult to avoid getting the impression that this paper also is about arthropod development and chitin remodeling. In my view it is not. However, combining the present data with literature data gives important novel insights into arthropod development, as pointed out in the Discussion section. In light of this, while the title is okay, some textual adjustments are needed elsewhere.

We appreciate the insight of the reviewer and have done our best to address the issues (see below).

Suggestions for changes:

(1) The abstract should be toned down (a little), in light of my comments.

We have toned down the abstract based on the reviewer`s comments.

(2) The present paper shows, for the first time, that a putative insect LPMO is active. This is very important, especially in the light of the –omics data which show that LPMOs are highly expressed and in the light of the presence of similar LPMOs in other higher eukaryotes. However, the fact that putative LPMOs occur in e.g. insects is not new; this has been predicted as early as in 2012 in a review by Horn et al., 2012 by simply searching genomes with the Pfam HMM (PF03067) (such a search yields Drosophila AA10s). CAZy is fantastic, and it is appropriate to create the AA14 class, but the reader deserves to know a little more about why CAZy creates several classes within the same Pfam entry. Also, in light of the above, the fact that this LPMO founds a new class in CAZy is as such not a sign of great novelty (biochemically, the characterized enzyme does not show great novelty neither).

I suggest the following adjustments:

2a. Previous predictions concerning the presence of LPMOs in insects and other higher eukaryotes should be mentioned and citations should be added.

We thank the reviewer for pointing this out. Indeed, putative LPMOs have been identified before in insects by computational analysis, although their activities and biological roles have never been elucidated. We have modified the paragraph “Phylogeny and sequence analysis” to acknowledge these facts.

2b. The rationale behind creating the AA14 class, and the relation between CAZy and Pfam needs to be explained better, in light of the comments above.

We have expanded paragraph “Phylogeny and sequence analysis” to describe the criteria that to define a new LPMO family through the use of HMMs (Hidden Markov Models). We have also included a supplementary table (Supplementary Data 5) showing the results of the HMMer search of 240 AA15 sequences against the HMMs of each LPMO family.

Although we do use Pfam for generic functional domain prediction, it does have some limitations, in particular the extremely degenerated profiles that are essentially motifs and do not cover the entire length of the catalytic domains. For this reason, we think Pfam is not suitable for classification/family assignment. CAZy, on the other hand, is based on HMMs built on whole sequence alignments and, importantly, is based on experimentally determined function for a founding member. Hence CAZy does not feature simply putative families, only biochemically determined ones (which is one of the main reasons why it is so widely adopted by the community).

2c. The authors should go through their manuscript and carefully consider the use of terms such as “new”, “unique”, in light of the above comments. Some specific suggestions:

- Lines 26-28 needs rephrasing (this sentence is suboptimal in several ways, including an overstatement of novelty; see also comment #3)

“new family” has now been changed to “uncharacterized family”. “to remodel” has been changed to “with possible roles in remodelling”.

- Lines 51-52 are a little ambiguous

Changed to “Until now, only LPMOs from bacterial, fungal or viral genomes have been characterized, with a predominant interest in their industrial applications towards bioethanol production.”

- Line 60: this family is uncharacterized in the CAZy world, which I happen to like a lot, but not in the Pfam world (where AA10 and at least some of the AA14s are in the same family, for which both chitin activity and cellulose activity have been shown, in 2010 and 2011, respectively).

As explained in response to comment 2b, we believe that Pfam is not suitable for classification and family assignment, mainly because it relies on highly degenerated profiles that do not cover the entire length of the catalytic domains. CAZy, we believe, is more appropriate as it uses HMMs covering the whole catalytic domain and is based on biochemically determined function for a founding member.

- Lines 85-90; I am quite sure that a simple Pfam search would have told you that there were putative LPMOs here.

The referee is correct. Pfam confirmed that the sequences contained a putative “LPMO_10” domain. In paragraph “Phylogeny and sequence analysis” we have added a note in brackets: “All sequences shared distant sequence similarity (between 20 and 30% amino acid identity) to lytic polysaccharide monooxygenases (as confirmed by Pfam search)...”

- Lines 93-95; here the use of the word “unexpectedly” is unacceptable.

The word “unexpectedly” has been deleted.

- Lines 100-102; significance comes in many forms.....

We apologise for the lack of detail. We have expanded the paragraph to describe the criteria used to define significance.

- Line 198: the structure does not unravel features related to promiscuity (see other comment), and this promiscuity is not “unusual”. One of the first cellulose-active AA10s (PF03067) that was ever structurally characterized also shows this promiscuity (Forsberg et al., 2014; PNAS, ScLPMO10B)

The first line of the paragraph has been re-phrased accordingly.

- Line 206-207: here you go..... (By the way, it would be appropriate to cite the CBP21 structure paper, which dates back to 2005 and presents the first structure of an LPMO)

We thank the reviewer for pointing out this reference, which has been duly added in the revised manuscript.

- Line 227: “entirely new”. Really? LPMO surfaces vary a lot.

We have rephrased the sentence as follows: “While having most of the canonical features found in other LMPOs, the AA15 structure reveals an unusual β -tongue-like protrusion which links strands 8 and 9 (Fig. 4a) and forms part of the surface surrounding the active site.”

- Lines 314-315

We have altered lines 314-315, which now read, “We present the first characterization of a new CAZy family of LPMOs (AA15) with roles in animal development and food digestion.”

- Line 339: delete “uniquely”

The word has been deleted as suggested.

- Line 432: delete “new”

The word has been deleted as suggested.

(3) The proteomics work is important and interesting, but I think it is wrong to say (or suggest) that this part of the study was essential for finding an active LPMO from an insect. Therefore, I suggest that the discovery aspect of the proteomics part of the manuscript is toned down a little. This is another reason to rephrase lines 26-28 (Abstract). Please make necessary minor adjustments throughout the manuscript.

Although we agree with the reviewer on the fact that there are various routes to discovering active LPMOs, the proteomics was actually the crucial experiment that kick started all the work here shown, while only later the rigorous classification of the whole AA15 family was performed using HMMs. The proteomics analysis highlighted the diversity of the LPMO pool (21 secreted enzymes) in the digestive system of *Thermobia*. Importantly, it also revealed the abundance of those enzymes and their likely role in plant biomass digestion. A simple bioinformatics analysis would not have provided these crucial elements. We have, however, rephrased lines 26-28 in the abstract to tone down our claims.

(4) In my view, as outlined above, there are no experiments in this study on arthropod development and “chitin remodeling”. Therefore, I suggest some rephrasing:

- Lines 64-66

The sentence has been toned down to “gene expression patterns and gene suppression phenotypes suggest that these ancient LPMOs play crucial roles in arthropod development and food digestion”

- The section starting on page 259 should have another title, and/or perhaps be split in two. The first two paragraphs of this section have nothing to do with deciphering the true biological function of the LPMOs. This is different for the third paragraph (and part of the Discussion section). I find lines 274 – 278 questionable; this is overselling as I see it. (Nb.

Line 273: what is the basis for the statement that termites “have not diversified their LPMOs” – How can we say that while we do not know the structural determinants of substrate specificity?)

We agree with the reviewer and have deleted those statements. We have also split the original paragraph in two (“Characterization of a chitin-specific LPMO from *Thermobia*” and “Possible role of AA15 LPMOs in chitin remodeling”).

(5) All MS pictures should be carefully checked for errors in peak labeling (e.g. Fig. 5c – incomplete labeling) and peak labeling should be made more clear (using helper lines?). Perhaps use an insert with zoom-in?

We did not detect the native (unoxidized) species for DP6, DP8 and DP10, hence the lack of labelling for those (not observed) peaks. This was always the case for activity assays with chitin (see Fig. 3b). The reason for the absence of the native species is not clear. However it could be related to the ability of the enzyme to recognize the repetitive unit (chitobiose) of the crystalline structure of the substrate (as mentioned in the legend of Fig. 3b), since even-numbered peaks were always more intense than the uneven-numbered ones. As suggested by the reviewer, we have added zoom-in insets in Fig. 3 a-b and Fig. 5 b-c.

Near line 165, the paper should contain an explicit statement concerning the possible occurrence of C4 oxidation. Could the native peaks visible in Fig. 3A be due to a C1/C4 mixed activity? Or could they be due to on-column degradation of C4 products, as pointed out by Westereng et al., 2016?

The reviewer is right in pointing out the possible issue of on-column degradation of C4 products. However, the MALDI analysis shown in our work was done on crude samples without any additional treatment (no column was used), therefore any C4 products would have been detected, had they been produced. Also, if there was substantial C4 oxidation, it would have also been seen in the permethylation experiment we performed (as it was the case for Quinlan *et al.* 2011). Based on these elements, we think there is strong evidence of C1 oxidation, but no evidence of C4 oxidation.

I am a little concerned when it comes to the MS spectra in Fig. 3 because the relative spacing between the -2, +16 and +38 products is not the same in panels A and B. I do not understand how that is possible.

The range of the x axis in Fig. 3a and 3b is different (500 to 2000 m/z for Fig. 3a, 500 to 2500 m/z for Fig. 3b) in order to accommodate all the major peaks in the two distinct experiments, carried out using Avicel and β -chitin as substrates, respectively. Hence the peaks in Fig. 3b appear closer to each other.

(6) Several Figures show synergy experiments based on single time point measurements. This is fine for the purpose of this study, but all quantification of these synergies must be deleted from the manuscript (including suppl material). LPMO kinetics are notoriously non-linear and many factors come into play that may not even relate to LPMO activity as such. True quantitative impressions of synergy may only be obtained by studying progress curves.

All quantifications from the synergy experiments have been deleted.

Other issues:

1. Lines 30-33 need some rephrasing. I do not really see how the “In-depth characterization” warrants the speculations in the second half of the sentence. Both are worth mentioning, but perhaps without the connection.

The sentence has been rephrased into “Based on our in-depth characterization of *Thermobia*’s LPMOs, we propose that diversification of these enzymes towards cellulose digestion might have endowed ancestral insects with an effective biochemical apparatus for biomass degradation, allowing the early colonization of land during the Paleozoic Era.”

*2. Line 113 & 124. It is extremely interesting that 21 LPMOs were detected, i.e. a very high fraction of the total number. This is not usually the case in proteomics studies of fungi, where a lesser fraction of the total numbers of LPMO tends to be detected. Does this really mean that a large number of the LPMOs is clearly expressed in *T. domestica*? Or is this due to e.g. another cut-off value? Perhaps some sort of quantitative statement could be made? Are all 21 clearly there?*

In our proteomics study, we used a particularly high stringency. The threshold for peptide spectral match acceptance was adjusted using Mascot Percolator to achieve a global false discovery rate of <1%. The false discovery rate is empirically estimated at the peptide level by comparing to a search against a decoy database using the same acceptance thresholds. The decoy database mimics the forward database in number of entries, amino acid distribution, protein lengths and cleavage site frequency but otherwise has randomized amino acid order. The achieved false discovery rate for this search was 0.59%, equating to 3,545 matches in the forward database and 21 in the decoy. The FDR estimation and significance filtering is at the peptide level so where LPMOs are matched through multiple peptides the confidence is greater still (for 17 out of 23 LPMOs the Mascot search detected two or more significant peptide matches, therefore reducing to a minimum the possibility of a false hit).

In light of this, it would also be interesting to know a little more about the RT-PCR experiments. How many of the 23 LPMOs were tested by RT-PCR and what were the results?

(Nb. I am not asking for more RT-PCR experiments, but if the authors have more information already, this should be included).

RT-PCR to compare expression in salivary gland, crop and midgut was carried out on three randomly selected LPMO sequences (shown in Supplementary Figure 5a), which all showed strong expression in the midgut. Although not included in the current study, we also used the midgut cDNA as template for four additional sequences (GASN01018175.1, GASN01000264.1, GASN01028018.1 and GASN01400743.1) and obtained good amplification for all those tested, although expression in the salivary glands and crop was not assessed for them (as the aim was to simply clone them). We also obtained good PCR amplification with all sequences tested (GASN01405718.1, GASN01018175.1, GASN01400743.1, GASN01404332.1 and GASN01030700.1) using total cDNA as template (from whole animal). So overall RT-PCR experiments indicate that LPMOs in *Thermobia* are clearly expressed, which is confirmed by the proteomics result. To further verify this hypothesis, we have retrieved the raw dataset published by 1KITE (accession: SRR921648), mapped the raw reads back onto the assembled contigs and calculated normalized expression levels (expressed as TPM, Transcripts Per kilobase Million) for all sequences (we have added a small section in Methods, in the RT-PCR paragraph). The results, included in a new table (Supplementary Table 1) confirm that most LPMO sequences are expressed at medium ($10 < \text{TPM} < 100$), high ($100 < \text{TPM} < 1000$) and very high ($\text{TPM} > 1000$) levels. The table is copied here:

Contig	Number of mapped reads	TPM
GASN01405718.1 = TdAA15A	49220	2040
GASN01030700.1	41936	1363
GASN01405900.1	19092	777
GASN02043256.1	20908	839
GASN01018175.1	19326	657
GASN01000262.1	11244	415
GASN01407914.1	10437	345
GASN01404332.1	5175	243
GASN01000264.1	2942	145
GASN01024742.1	2475	129
GASN01028566.1	1285	78
GASN01400743.1	1198	73
GASN01404421.1	1491	69
GASN01009794.1	1149	66
GASN01404396.1	728	34
GASN01028018.1	410	22
GASN01011237.1	552	22
GASN01028062.1	300	18
GASN01024272.1	303	13
GASN01405771.1	136	6
GASN01404868.1	74	3
GASN01010363.1 = TdAA15B	74	1
GASN01008505.1	41	1

Interestingly, the sequence coding *TdAA15A* is the most highly expressed among all LPMOs from *Thermobia*, and is among the top 20 most expressed genes of the whole transcriptome (see table below, for reviewers only):

Rank	Contig	NumReads	TPM
1	GASN01371451.1	215708	36868
2	GASN01310188.1	24880	32358.4
3	GASN01005933.1	46833.9	10198.3
4	GASN01375815.1	54810	8504
5	GASN01409931.1	290331	7217.85
6	GASN01279101.1	3722	5939.06
7	GASN01410534.1	199138	4451.13
8	GASN01383886.1	34844	4442.15
9	GASN01001342.1	31203	4304.18
10	GASN01371059.1	22550	4233.6
11	GASN01021578.1	23701	3738.16
12	GASN01382496.1	25876	3436.75
13	GASN01000646.1	70543	2837.44
14	GASN01335322.1	6254	2822.73
15	GASN01026157.1	27343	2569.68
16	GASN01354822.1	9142	2286.63
17	GASN01020589.1	172414	2237.93
18	GASN01405718.1 = TdAA15A	49220	2040.03
19	GASN01013883.1	31072	2013.27
20	GASN01032069.1	264749	2000.01

3. Line 120-122. How well was methylation assessed (for example, for how many of the LPMOs was the N-terminal peptide actually detected in the proteomics study?)

When we searched tandem mass spectra against the whole transcriptome of *Thermobia* (which included the full protein sequences before any downstream processing, such as signal peptide removal), we did not detect any masses compatible with the presence of the N-terminal peptide for any of the LPMOs. However, as mentioned in the Methods of the first version of the manuscript (lines 421-422), we also set up a specific search in Mascot using histidine methylation as a variable parameter and interrogated the pool of mature LPMO proteins (without N-terminal peptide), which detected only masses compatible with a non-methylated histidine at the N-terminus. These results indicate that: 1) mature LPMOs lack the N-terminal peptide; 2) mature LPMOs have a non-methylated N-terminal histidine. We have now included more details in the Methods of the revised manuscript (paragraph “Shotgun proteomics”).

4. Line 210 and below: It would be nice to explain to the reader what is meant/implied by “two species”.

We have clarified the statement by adding further information in the discussion of the EPR experiments and further supplementary figures (Supplementary Figures 10 and 11).

5. Line 224. It would be appropriate to cite Kracher et al., 2016.

We have now added the suggested reference.

6. 328-329: Another selling statement. How could this work? Lines 329-337 could be deleted; this is speculation (how could this work?)

We appreciate the reviewer's comment and concur that this section is perhaps too speculative, so we have deleted it.

7. Line 469-472: Any reason for not using gene synthesis?

When carrying out heterologous expression, we routinely test the native sequences first, since codon optimised genes are much more expensive. Only when expression of native sequences fails, we switch to gene synthesis. In this case, *Thermobita*'s native LPMO sequences could be successfully expressed in *E. coli*'s periplasm in an active form and with good yield.

8. Line 473-475: Unclear; add more details.

As suggested, we have expanded this section with additional details.

9. Line 502 and onwards: Where no protease inhibitors used?

Apologies for the lack of detail. We used AEBSF as protease inhibitor (now added to the methods).

10. Fig. S3: Please indicate the protrusion.

We have now amended the picture as suggested and highlighted the protrusion (166-YGDCGDGTSGMGC-178).

11. Legend to Fig. S5: which LPMOs were tested by RT-PCR? Does the list include TdAA14A? Please indicate.

In Supplementary Fig 5a, lane a corresponds to TdAA15A (sequence GASN01405718.1). The legend has now been corrected to clarify this.

12. Legend to Fig. 4c: Are these four LPMOs C1, C4 or C1&C4 oxidizers? I would suggest citing the papers behind each of these structures.

We welcome this useful suggestion. Citations to each structure have been added in the figure legend and we have also highlighted the regio-selectivity and substrate specificity of each enzyme.

Reviewer #3 (Remarks to the Author):

*This paper describes the cloning, expression, protein purification, and kinetic, structural, and EPR characterizations of a recombinant lytic polysaccharide monooxygenases (LPMO) present in the digestive proteome of *Thermobia domestica*. There are several examples of LPMOs reported in the literature, the novelty of the paper is that the protein characterized represents the first LPMO from an animal genome.*

I have some concerns with the MS that should be addressed before publication in Nature comms.

1) On pages 7-9. Biochemical characterization. The role of copper in the catalytic mechanism should be better explained. Copper is incorporated into the apoprotein on addition of copper excess. Apparently, the apoprotein is not catalytically competent; however, it is not clear whether the catalytic activity of the apoprotein was checked or not.

As the reviewer has pointed out, copper is crucial for the activity of LPMOs, and AA15s are no exception. To verify the lack of activity by the apo-protein, we have carried out *in vitro* activity assays with both *TdAA15A* and *TdAA15B* using the same experimental conditions shown in Fig. 3a-b and Fig. 5b-c, in presence of 10 mM EDTA. MALDI-TOF MS analysis of the crude supernatant did not reveal the formation of native nor oxidized oligosaccharides, indicating that the apo-enzymes were completely inactive. We have added the following sentence (paragraph “Biochemical characterization”): “MALDI-TOF MS analysis of crude extract from activity assays carried out with Cu-loaded *TdAA15A* in presence of 10 mM EDTA failed to detect the release of both native and oxidized oligosaccharides (Supplementary Fig. 7 a,b), indicating that the apo-enzyme is not active and that copper is essential for activity.” Similarly, we have now mentioned the result of the assay with *TdAA15B* in paragraph “Characterization of a chitin-specific LPMO in *Thermobia*”: “MALDI-TOF MS analysis of crude extract from activity assays carried out with *TdAA15B* in presence of 10 mM EDTA failed to detect the release of both native or oxidized oligosaccharides from both α and β -chitin (Supplementary Fig. 13 a,b), indicating that copper is crucial in activating the enzyme (as previously observed for *TdAA15A*).” The spectra from reactions of both enzymes with EDTA been added as Supplementary Figures 7 and 13.

Copper content and copper/protein ratio of reconstituted LPMO is not reported, which should be included in the MS.

In Methods, paragraph “cDNA cloning, heterologous expression and purification of recombinant *TdAA15A* and *TdAA15B*” states “5 fold excess copper was added as CuSO_4 , then unbound copper and desthiobiotin were removed by passing the protein in a HiLoad™ 16/60 Superdex 75 gel filtration column (Ge Healthcare) equilibrated with 10 mM sodium phosphate buffer pH 7.”

To answer the question regarding the copper/protein ratio of reconstituted LPMO, we have carried out ICP-MS analysis on *TdAA15A* (after copper loading and gel filtration) and obtained a value of 1.1 ± 0.2 , thus indicating close-to-equimolar amounts of copper and protein. We have added this result in the paragraph “Biochemical characterization” and a new Methods paragraph.

The redox role of the copper center in catalysis, which is slightly mentioned in the MS, should be better explained. Authors provide experimental evidences highlighting the structural role played by copper but only a brief mention on the copper redox role is given. Although a complete redox characterization should be desirable, the author should at least provide EPR information of the copper centers by reporting the copper behavior towards conventional reductants such as ascorbate and dithionite and oxidants such as ferricyanide. This standard procedure should give at least information of the order of magnitude of the copper reduction potential and perhaps helps to discriminate the EPR signals associated with the two copper species that supposedly are present in LPMO (see below also). The EPR behavior towards the electron donors used in the catalytic assays should be included and analyzed. The UV-vis properties, mainly those associated with the copper centers, should also be reported.

We thank the reviewer for his/her comments. We have carried out a more detailed spectroscopic characterization of *TdAA15A*, now part of a separate paragraph (“Spectroscopic features of *TdAA15A*”). EPR experiments in the presence of ascorbic acid or ferricyanide have been performed and added to the manuscript (Supplementary Figure 10).

Supplementary Figure 10 | Continuous wave X-band EPR spectra (9.3 GHz, 160 K) of *TdAA15A*. EPR spectra of 0.3 mM *TdAA15A* in a mixed buffer composed by sodium acetate, MES, HEPES and TRIS (10 mM each) at pH 6 (a), pH 8.5 (b) or pH 6 with 10% glycerol added (c), with simulations shown in red. Simulations were obtained with 85% of species 1 and 15% of species 2 in (a), 100% of species 1 in (b) and 45% of species 1 and 55% of species 2 in (c). See Supplementary Table 3 for details of spin Hamiltonian parameters. (d) EPR spectrum of 0.2 mM *TdAA15A* in 20 mM sodium phosphate buffer pH 6 recorded before (black) or 10 min after addition of 100 equivalents of the reductant sodium ascorbate (red). (e) EPR spectrum of 0.2 mM *TdAA15A* in 20 mM sodium phosphate buffer pH 6 with 10% glycerol recorded before (black) or 10 min after addition of 20 equivalents of the oxidant potassium ferricyanide (red). The signal at ~3320 G in (d) and (e) is due to the EPR tube.

The UV-vis spectrum of the protein at 1 mM concentration was also collected and shows, as expected for this kind of copper centre, a very broad feature with a molar extinction coefficient $\epsilon = 75 \text{ M}^{-1} \text{ cm}^{-1}$, typical for d-d transitions (Supplementary Figure 9). A more detailed assignment of the transitions is not possible without further spectroscopic experiments, which are beyond the scope of this paper. The UV-vis data have been added to the manuscript as Supplementary Figure 9.

Supplementary Figure 9 | UV/vis spectrum of TdAA15A. Spectrum was collected with 1 mM protein in 20 mM sodium phosphate buffer pH 7.

2) On page 10. Structural and spectroscopic characterization.

The EPR spectrum of LPMO is interpreted on the basis of two superposed spectral components with different EPR parameters. Superhyperfine interactions with the coordinating ligands is also included, but one has to go the supplementary material to understand the latter.

A sentence has been added to the EPR methods to clarify that coordinated nitrogen atoms are included in the simulations.

The simulation assumptions are a matter of criticism. Firstly, one could suppose two spectral components to simulate the spectrum, but not to include N superhyperfine interactions, as no experimental evidences can be detected in the spectrum. Furthermore, isotropic N-couplings (Table S2) do not have physical sense. The assumption of two spectral components from the single EPR spectrum reported is rather controversial, as additional EPR experiments should be necessary to obtain such a conclusion. There are several possible explanations for the EPR spectrum of LPMO. Note that the X-ray crystal data, which are taken with 1.1 Å resolution, show only a single metal center with no evidences of disorder, in which case the

simplest interpretation for the EPR experiment in solution is some metal site inhomogeneity. I think that the number of such conformations cannot be assessed with the present data. In other words, the author's proposal would require additional conventional EPR experiments such as EPR redox titration complemented with spin quantifications, and more specialized ones such as EPR at different MW frequencies and N-hyperfine interaction detection by more sophisticated magnetic resonance techniques.

Direct experimental evidence of SHF coupling can be detected in the second derivative spectra, and the presence of coordinated nitrogen atoms has an influence on the overall aspect of the EPR, in particular in the shape of the perpendicular region. Satisfactory simulations could only be obtained by addition of three nitrogen atoms for species 1 and two nitrogen atoms for species 2. Furthermore, well resolved SHF coupling is visible in the EPR spectrum collected at pH 8.5 (Supplementary Figure 10), confirming the presence of coupled nitrogen atoms with SHF coupling values as reported in Supplementary Table 3. The second derivative spectrum of the EPR at pH 8.5 is shown below for the reviewer, with experimental data in blue and simulations in red. The experiments and the results of the simulations have been added to the EPR methods and the Supplementary Informations.

Second derivative EPR spectra of *TdAA15A* at pH 8.5. Full spectrum (left) and detail of the perpendicular region (right), with simulations shown in red.

As suggested by the reviewer, we performed further EPR experiments, now included in the Supplementary Information, to evaluate the presence of two species in solution. These EPR pH titrations in the presence or absence of 10% glycerol confirmed the initial observations described in the study, with one, more axial, species being predominant in the absence of glycerol, and a second, more rhombic, one appearing in higher amount when glycerol was added to the sample (Supplementary Figure 11).

Supplementary Figure 11 | Continuous wave X-band EPR pH titration spectra (9.3 GHz, 160 K) of *TdAA15A*. EPR spectra of 0.3 mM *TdAA15A* in a mixed buffer composed by sodium acetate, MES, HEPES and TRIS (10 mM each) without (a) and with (b) 10% glycerol. Spectra recorded every 0.5 pH units between pH 5 and pH 8.5. For clarity, only a selection of spectra is presented; full data are available through the Research Data York (DOI: 10.15124/bd09e86b-9d92-4802-9337-18b138e7abb7).

We agree with the reviewer that there is only a single metal centre in the structure of the protein. The crystal structure obtained, although at very high resolution, presents the metal centre in the Cu(I) oxidation state, which would retain a three-coordinate configuration, therefore giving no information about differences in the coordinated ligands in the Cu(II) state. For these reasons, as the reviewer also suggests, we believe that the EPR behaviour can convincingly be explained by the presence of two species which differ in their coordinated exogenous ligand. In particular, the spin Hamiltonian parameters for the more rhombic species (species 2) would suggest a certain degree of d_z^2 mixing into the SOMO, consistent with coordination of glycerol or buffer to the Cu ion.

We agree that even further EPR experiments, such as multi-frequency continuous wave EPR or pulsed experiments, would be beneficial for the understanding of the detailed electronic aspects of the active site, but we feel that they would go beyond the scope of this manuscript, which is focused on the identification and characterization of a new LPMO family. Such an in-depth spectroscopic study as the review suggests, currently on-going in our laboratories, will be the body of a separate publication.

MS and supporting information. “Accurate determination of perpendicular values and superhyperfine coupling constants was not possible due to second order nature of the spectrum in this region” Please, clarify the meaning of “second order nature etc” Do authors want to mean second order corrections usually employed in perturbation theory?

‘Second-order’ is a magnetic spectroscopy term used to describe spectra with overlapping and unresolved features.

Fig. 4, d. Simulation of each spectral component should be included.

We have now added the simulation for the individual components of the spectra (Supplementary Figure 10).

On page 10, lines 2017-210. “While the third, non-coordinating active site residue of TdAA14A is a tyrosine (Tyr184) as in most AA9s, the positioning of Ala89 is reminiscent of AA10s (Fig. 4c) and such features are reflected in the rhombic EPR spectrum of the Cu(II) form of TdAA14A (Fig. 4d, Supplementary Table 2)”

We thank the reviewer for pointing out the lack of clarity in our statement. The effect on the Cu ion is due to the presence of Ala89 rather than the Tyr184. The alanine residue can provide steric hindrance for the coordination of water molecules around the metal, hence distorting the geometry and influencing the spin Hamiltonian parameters. The distortion of the geometry from square planar to trigonal bipyramid is well documented for AA10s (e.g. Gudmundsson, M. *et al. JBC*, **289**, 18782-18792 (2014) and Gregory *et al. Dalton Trans.*, 2016,**45**, 16904-16912). We have amended the sentence to eliminate the confusion.

It sounds strange that non-coordinating copper ligands determine the symmetry of the Cu(II) g-matrix. Note also the copper as depicted in Fig. is 3-coordinated. This is rather unusual for a Cu(II) ion, which deserves some comment.

We apologise for the lack of clarity, which perhaps caused some confusion here. The copper ion in the structure is Cu(I), hence the three-coordinate state. We have updated the figure legend to make this clearer to the reader.

On page 10, lines 212-213 and 216-219. “The mixed species observed and the variation in their ratio in response to exogenous ligands and pH reveal” These EPR data should be shown either in the MS or as supporting information.

We have added the spectra obtained from the pH titrations and the spectra with corresponding simulations for the two species to the supplementary information.

Note also that glycerol was added to the EPR samples as cryoprotectant. One should avoid this very old procedure when possible, as glycerol is a potential Cu(II) ion ligand. EPR studies with and without glycerol should be analyzed and compared.

We agree with the reviewer that glycerol can be, as in the current study, a potential ligand for the Cu(II) ion and for that reason we check the binding by performing experiments in the presence or absence of this cryoprotectant. If differences are detected, we avoid the addition of glycerol. The EPR investigation with and without glycerol has been carried out and added to the manuscript.

Reviewers' Comments:

Reviewer #1:

Remarks to the Author:

N/A

Reviewer #2:

Remarks to the Author:

The authors have done a good job when revising this paper, the eventual publication of which I warmly recommend. Some of the changes are really nice, such as the better explanation of how and why the new CAZY family was established.

I have two minor and one not so minor comment:

Minor:

- Line 345: I suggest to change "with roles in" to "with putative roles in"
- Terms like "unique" and "unexpectedly" have been removed, for good reasons, and I am happy with that. I think, however that it would be correct to explicitly point out that LPMOs that are active on both chitin and cellulose are known from previous work. This could for example be done by adjusting the legend of Fig. 4 (line 273). SclPMO10B is an example of such an LPMO with dual chitin and cellulose activity, and its activity on chitin could (should) be mentioned here. (This minor comment is distantly related to my not so minor comment; perhaps some more adjustment would be justified).

Less minor:

Lines 304-306 ("Our protein sequence analysis....") are the remainder of a longer section which I suggested to change when reading the first version of this manuscript. Unfortunately, the revised version enhances, rather than reduces the problem that I had with the original text. The revised text suggests even stronger than before that the presence of an axial tyrosine or an axial phenylalanine is a major determinant of substrate specificity (cellulose or chitin, respectively). In my view, there is no basis for claiming this, especially not since one of the best known C1-oxidizing cellulose-active (and not chitin-active) AA10s has an axial phenylalanine (SclPMO10C or CelS2). Fact is that the structural determinants of substrate specificity are not known and this should somehow be acknowledged in the text. It is fine if the authors wish to note the correlation between the presence of an axial Tyr and a tendency to be active on cellulose, but strong claims should be avoided and the occurrence of known exceptions should be acknowledged. When adjusting the text, the word "unique" in line 304 should be deleted.

Reviewer #3:

Remarks to the Author:

The authors were responsive to my suggestions, providing some alterations in the revised manuscript. However, there are some minor issues that in my opinion still need to be clarified in the MS.

Supplementary Fig. 10. Authors simulate the spectra "a" and "b" with two and one components, respectively. I don't see any significant difference in the experimental spectra justifying such a procedure. Perhaps there are some details I cannot appreciate but as far as I can see spectra "a" and "b" are in fact the same.

Spectrum "c". Undoubtedly, glycerol addition is responsible for the second species. Hence, my

question is why to insist with glycerol considering that this molecule severely disturbs the Cu(II) site. The changes introduced by glycerol in the EPR spectra should be commented in text. As stated in my original review, spectral components associated with species 1 and 2 should be included, which should be helpful to assess how the authors obtain the full simulation (only the simulation parameters are given)

EPR spectra "d" and "e" in Supp. Fig. 10 were obtained with a different buffer, why? The results allow the authors to conclude that reduction potential of the Cu(II) species fall in the range 0-200 mV vs NHE, which should be clarified in the text. Check the amount of added ferricyanide (100 equivalents?). Again, I don't understand why the studies were performed in the presence of glycerol taking into account the above comments.

Author's response: "Second-order' is a magnetic spectroscopy term used to describe spectra with overlapping and unresolved features"

I still think that second order is not an appropriate term. Please, try to simplify and think that this paper should be directed to a broad audience.

Despite authors provide some experimental evidences the N-hyperfine structure is solved at high pH, this structure is not resolved at low pH. One should be more cautious with these interpretations since I think that this structure is included based solely on structural data.

No EPR spectra with enzyme reductants such as gallic acid was reported. Note that gallic acid is an important enzyme electron donor.

Reviewer #2 (Remarks to the Author):

- Line 345: I suggest to change “with roles in” to “with putative roles in”.

The sentence has been changed as suggested.

- Terms like “unique” and “unexpectedly” have been removed, for good reasons, and I am happy with that. I think, however that it would be correct to explicitly point out that LPMOs that are active on both chitin and cellulose are known from previous work. This could for example be done by adjusting the legend of Fig. 4 (line 273). ScLPMO10B is an example of such an LPMO with dual chitin and cellulose activity, and its activity on chitin could (should) be mentioned here. (This minor comment is distantly related to my not so minor comment; perhaps some more adjustment would be justified).

We have amended the sentence in the figure legend.

Lines 304-306 (“Our protein sequence analysis.....”) are the remainder of a longer section which I suggested to change when reading the first version of this manuscript. Unfortunately, the revised version enhances, rather than reduces the problem that I had with the original text. The revised text suggests even stronger than before that the presence of an axial tyrosine or an axial phenylalanine is a major determinant of substrate specificity (cellulose or chitin, respectively). In my view, there is no basis for claiming this, especially not since one of the best known Cl-oxidizing cellulose-active (and not chitin-active) AA10s has an axial phenylalanine (ScLPMO10C or CelS2). Fact is that the structural determinants of substrate specificity are not known and this should somehow be acknowledged in the text. It is fine if the authors wish to note the correlation between the presence of an axial Tyr and a tendency to be active on cellulose, but strong claims should be avoided and the occurrence of known exceptions should be acknowledged. When adjusting the text, the word “unquie” in line 304 should be deleted.

Lines 304-306 have been deleted as suggested.

Reviewer #3 (Remarks to the Author):

Supplementary Fig. 10. Authors simulate the spectra “a” and “b” with two and one components, respectively. I don’t see any significant difference in the experimental spectra justifying such a procedure. Perhaps there are some details I cannot appreciate but as far as I can see spectra “a” and “b” are in fact the same.

We agree that it's difficult to see by eye any major differences, however we are unable to simulate spectrum "a" with a single species. In particular, there is a shoulder at about 2950 G in panel "a" that is not present in panel "b", and there are also some changes in the perpendicular region. Most importantly however, titration (Supplementary Fig. 11a) shows the disappearance of species 2, showing that it is related to a chemical event. The context of this analysis is also important insofar as only 15% of the second species is present at pH 6, so this does not lead to an ostensible significant change in the appearance of the spectrum, but the change is certainly significant at the level of simulation and should therefore be included in any analysis.

Spectrum "c". Undoubtedly, glycerol addition is responsible for the second species. Hence, my question is why to insist with glycerol considering that this molecule severely disturbs the Cu(II) site. The changes introduced by glycerol in the EPR spectra should be commented in text.

We respectfully disagree with this comment, the second species is present also in the spectra without glycerol (Suppl Fig.11a) and we observe it in spectra with no glycerol but different buffers (see below for a comparison between MES and phosphate). By addition of glycerol we are illustrating that the active site of this particular protein is particularly responsive to different ligands. All of this is already clearly stated in the manuscript, viz "their ratio (of the two species) was shown to be dependent on pH, buffer and glycerol content." and we do not believe that this point requires further elaboration.

A stated y my original review, spectral components associated with species 1 and 2 should be included, which should be helpful to assess how the authors obtain the full simulation (only the sim parameters are given)

We show the spectrum for the sample with 100% species 1. We are unable to obtain a

sample with 100% species 2, but once species 1 is simulated within the spectrum of the mixed sample, we can obtain the spectrum of species 2 by subtraction, (which of course we know from the sample with 100% species 1).

EPR spectra “d” and “e” in Supp. Fig. 10 were obtained with a different buffer, why? The results allow the authors to conclude that reduction potential of the Cu(II) species fall in the range 0-200 mV vs NHE, which should be clarified in the text.

The buffer is the same one as used in the activity assays. When ferricyanide is added, the EPR signal is completely suppressed if glycerol is not present, hence the difference between the two samples.

We are reluctant to estimate a reduction potential for the LPMO using this procedure, since the differing concentrations of the reducing agent and its oxidised form mean that the onset reduction potential is significantly different from that of the published reduction potential of the oxidised form of the reducing agent (i.e. we are at non-Nernstian conditions)

Check the amount of added ferricyanide (100 equivalents?). Again, I don't understand why the studies were performed in the presence of glycerol taking into account the above comments.

We believe that the reviewer must have misread as the amounts of ferricyanide are clearly stated in the EPR methods section, ("Similarly, **20 equivalents** of potassium ferricyanide were added to a 0.2 mM sample of TdAA15A"). As reported in the above comment, glycerol is necessary in the presence of ferricyanide to obtain an EPR signal.

Author's response: "Second-order" is a magnetic spectroscopy term used to describe spectra with overlapping and unresolved features" I still think that second order is not an appropriate term. Please, try to simplify and think that this paper should be directed to a broad audience.

We have removed the phrase 'second order'.

Despite authors provide some experimental evidences the N-hyperfine structure is solved at high pH, this structure is not resolved at low pH. One should be more cautious with these interpretations since I think that this structure is included based solely on structural data.

I am afraid that we respectfully disagree with the reviewer's opinion. It is simply a fact that the simulation requires the presence of nitrogen atoms such that the perpendicular region is simulated in a satisfactory way--the inclusion is **not** solely on the basis of the structural information. We believe that this is the best practice for EPR simulations. Furthermore, in this case, we can actually resolve the nitrogen features in the spectrum at pH 8.5. They are

not resolved at low pH probably for several reasons, one of which being the presence of the second species which broadens the appearance of the perpendicular region (even if present as only 15% of the total, see above), but they are still required for a successful simulation.

No EPR spectra with enzyme reductants such as gallic acid was reported. Note that gallic acid is an important enzyme electron donor.

Within the EPR spectrum addition of gallic acid does not appear to reduce the copper. We have added a note to this effect in the methods section (see below). At this stage we direct the reviewer to the recent publication by Kracher *et al.* (PMID 29259126) who demonstrate that there is a subtle interplay between the oxidation state of the Cu and the presence of substrate. The fact that gallic acid does not appear to reduce the Cu in the absence of substrate does not mean to say that it cannot be the active reducing agent. Indeed, it may be beneficial for the LPMO only to be reduced in the presence of substrate, as our experiments with gallic acid would seem to suggest.

The following sentence has been added to the methods: "Gallic acid was added to the protein in the same conditions, but no decrease in the copper signal was visible over a 3 h period, which- since gallic acid is clearly active in the assays -implies that the reduction potential of the copper active site is affected by the presence of substrate."